**EMBO** *reports*

# Duox-driven ROS release by glia promotes regeneration in the adult Drosophila brain

Carolina S Alves[1], Anabel R Simões[1], Beatriz Gil Ferreira[1], Marta Neto[2], Carmo C Soares [ID][1],
Andreia Augusto[1] & Christa Rhiner [ID][1✉]

## Abstract

Tissue damage activates immediate responses to restrict further harm and initiate repair. How injury sensing is coupled to regeneration is still not well understood. Here, we study regenerative responses in the adult *Drosophila* brain, where proliferation is normally strongly restricted. We show that localized brain damage triggers oxidative stress and diverse brain protective programs. We find that ROS generation by the NADPH Oxidase Duox in glial cells is responsible for injury-induced oxidative stress. Both genetic and chemical suppression of ROS in injured brains impairs regeneration. In particular, selective knockdown of calcium-sensitive Duox in glia, which show elevated calcium after injury, reduces injury-induced proliferation. We further provide evidence that diffusing ROS can sustain the activity of pro-regenerative signaling, which is required to stimulate cell divisions. Although oxidative stress is generally considered as harmful in the brain, we uncover here an unanticipated beneficial role of transient ROS release by glia to promote brain repair.

**Keywords** Brain Plasticity; Glia; ROS; Duox; *Drosophila*
**Subject Categories** Metabolism; Neuroscience

## Introduction

Tissue injury represents a serious threat to the health and survival of an organism. Hence, animals have evolved highly conserved protective responses to contain damage and promote repair. These include the activation of immune responses, debris clearing, extracellular matrix remodeling, and cell proliferation to restore tissues.

Tissue damage also causes oxidative stress in nearby cells, which can play a dual role in signaling or as a damaging agent depending on levels, temporal dynamics, and tissue-specific effects, which are still to be uncovered.

Small changes in free radicals regulate important aspects of normal growth factor signaling, nervous system development, and stem cell maintenance (Schieber and Chandel, 2014; Oswald et al, 2018). Strong and perduring ROS elevation leads to engagement of cytoprotective programs via transcription factors such as Nrf2 and AP1 (Weavers et al, 2019; Sykiotis and Bohmann, 2008).

Under physiologic conditions, the majority of free radicals arise from mitochondrial respiration; another large fraction is generated by NADPH oxidases (Duox and Nox proteins), which produce ROS at the extracellular face of cell membranes (Sies et al, 2022). Roles of ROS-releasing NADPH oxidases have been successfully elucidated in fruit flies: ROS-bursts generated by immune cells function to fight off pathogens (Ha et al, 2005), whereas gradients of ROS serve as chemoattractive cues that recruit immune cells to epithelial wounds (Razzell et al, 2013; Moreira et al, 2010), as initially found in Zebrafish (Niethammer et al, 2009). Additionally, ROS have been shown to modulate signaling to promote growth of damaged epithelia during development (Santabárbara-Ruiz et al, 2015; Khan et al, 2017; Brock et al, 2017).

In contrast to their function in immune defense and epithelial repair, the role of ROS related to brain damage, especially acute injury, is less well understood. For neurodegenerative disease, ROS and mitochondrial dysfunction have been identified as key causative factors (Lin and Beal, 2006), and mitochondrial dysfunction and lasting oxidative stress promote disease progression in fly models of Parkinson's and Alzheimer's (Bou Dib et al, 2014; Moulton et al, 2021). Peripheral nervous system injury and epidermal damage (fin amputation) in zebrafish revealed a dual role for NADPH oxidases, with Nox causing sensory axon degeneration and Duox facilitating peripheral nerve regeneration via EGFR-dependent ECM remodeling (Hervera et al, 2018). Moreover, an immune cell provided Duox function promoted nerve regeneration in the mouse peripheral nervous system (Cadiz Diaz et al, 2022), but the impact of oxidative stress on brain cell interactions still remains elusive.

Here, we study localized brain injury in the adult fly brain and find that specific ROS released by glial cells can act as an important pro-regenerative signal.

## Results and discussion

### Injury signals and tissue protective pathways

Acute damage to the fly brain or ventral nerve cord trigger conserved repair responses including glial reactivity, glial

[1]Champalimaud Foundation, Champalimaud Research, Lisbon 1400-038, Portugal. [2]Instituto de Neurociencias, Consejo Superior de Investigaciones Científicas (CSIC) and Universidad Miguel Hernández (UMH), San Juan de Alicante 03550, Spain. ✉E-mail: christa.rhiner@research.fchampalimaud.org

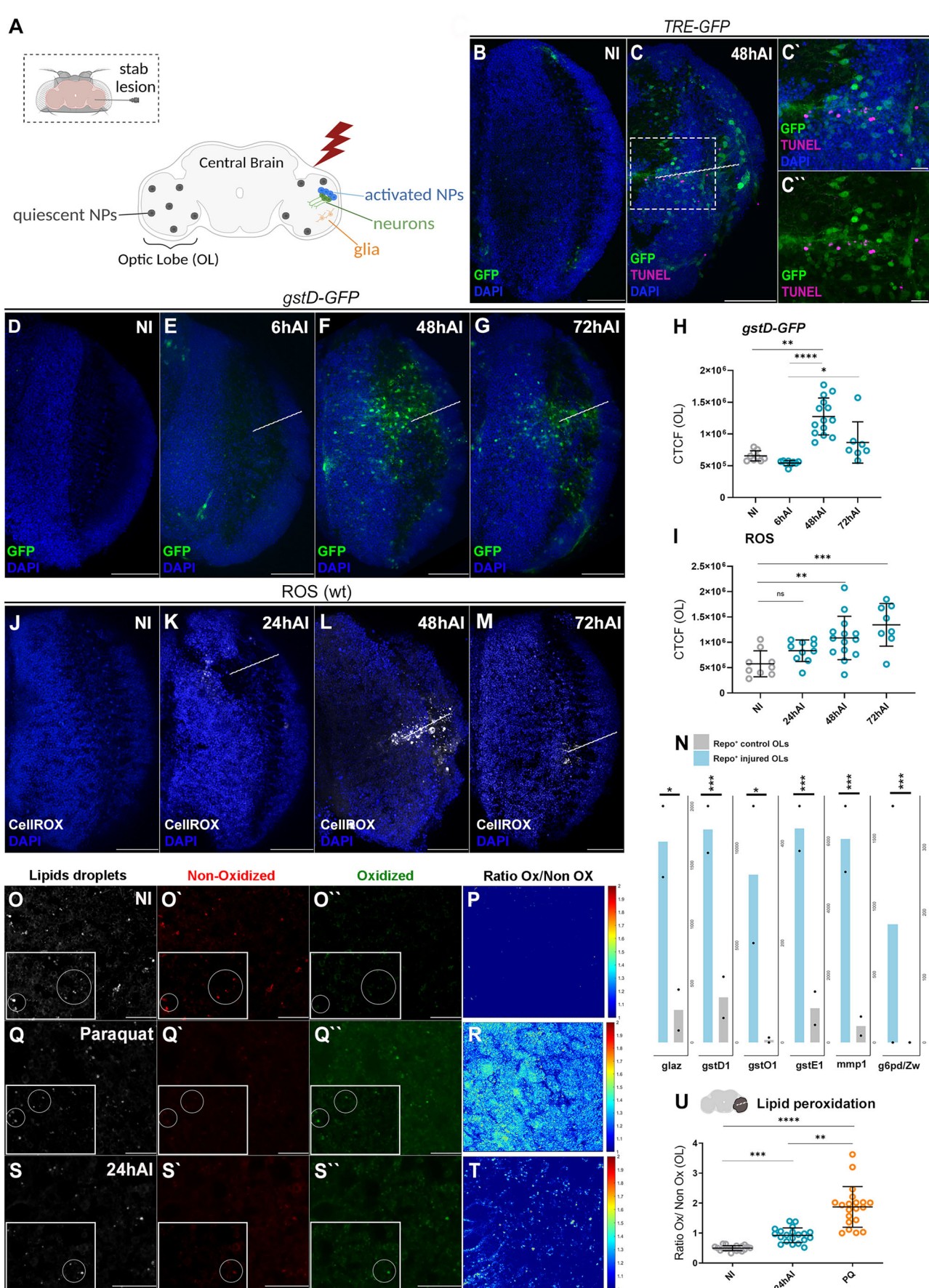

◄  **Figure 1.  Brain lesions induce ROS and protective antioxidant responses.**

(A) Schematic depicting stab lesions to the optic lobe (OL) of the adult fly brain. The injury triggers the activation of normally quiescent neural progenitors (NPs) and glial divisions. (B, C) Control (B) and injured OL showing apoptotic cells (TUNEL, magenta) and JNK activation (*TRE-GFP*, green) 48 h after injury (AI) with insets for the boxed area (C). Dashed white lines indicate stab lesion in all subsequent images. Nuclei are counterstained with DAPI (blue). (D–G) Oxidative stress levels (*gstD*-GFP) AI. (H) *gstD*-GFP fluorescence intensity in non-injured (NI) and injured OLs (AI). Data is shown as mean ± SD. ****$P < 0.0001$, **$P < 0.01$ ($P = 0.0049$), *$P < 0.05$ ($P = 0.025$) by nonparametric Kruskal–Wallis test with Dunn's multiple comparison correction $N = 9$ (NI), 9 (6 h) 14 (48 h) 7 (72 h) OLs. CTCF corrected total cell fluorescence. (I) ROS quantification (CellROX). Data is shown as mean ± SD. ***$P < 0.001$ ($P = 0.0003$), **$P < 0.0084$, n.s. ($P = 0.39$) by one-way ANOVA with Tukey's multiple comparison. $N = 9$ (NI), 10 (24 h), 14 (48 h), 8 (72 h). (J-M) ROS (CellROX, white) AI. Nuclei are shown in blue (DAPI). (N) Upregulated genes in glia from injured (blue) and control OLs (gray). Padj based on Benjamini–Hochberg False Discovery Rate ***$P < 0.001$, **$P < 0.01$, *$P < 0.05$ ($P = 0.033$, *glaz*; $P = 0.0009$, *gstD1*; $P = 0.017$, *gstO1* $P = 0.00022$, *gstE1*; $P = 0.00068$, *mmp1*; $P = 3.51E-10$, *g6pd*). Y axis refers to normalized read counts. (O) CNS lipid peroxidation with PUFA ratiometric peroxidation sensor C11-BODIPY 581/591 and neutral lipid stain LipidTox in non-injured (NI) optic lobes (OLs). Boxed: Lipid droplet insets. (P) Ratio of oxidized vs non-oxidized lipids in control OLs. (Q) Elevated lipid oxidation in OLs upon paraquat treatment. (R) Increased oxidation ratio in paraquat-exposed OLs. (S) Lipid peroxidation in injured OLs, 24 h AI. (T) Ratio of oxidized vs non-oxidized lipids in OL area targeted by the stab lesion. (U) Lipid droplet peroxidation in the OL (ratios Ox/non-Ox). NI non-injured, AI after injury, PQ paraquat-treated. Data are shown as mean ± SD. ****$P < 0.0001$, ***$P < 0.001$ ($P = 0.0008$), **$P < 0.01$ ($P = 0.0035$) by nonparametric Kruskal–Wallis test with Dunn's multiple comparison correction. $N = 20$ for all conditions. Scale bars are 50 μm for OL images, 20 μm for lipid images, and 10 μm for the insets. Source data are available online for this figure.

proliferation and the activation of normally quiescent neural progenitor cells that can give rise to new neurons (Lu et al, 2017; Kato et al, 2009; Fernández-Hernández et al, 2013; Crocker et al, 2021; Simões et al, 2022; Casas-Tinto et al, 2025). Increased glial proliferation at the wound site is also observed in the developing ventral nerve cord (Kato et al, 2011). The acute damage inflicted on the nervous system activates conserved pro-regenerative signaling such as JNK, Wg/Wnt, and insulin (Simões et al, 2022; Saikumar et al, 2020; Harrison et al, 2021). In the adult optic lobe, acute brain damage by stab lesion triggers localized cell death, JNK signaling, and the proliferation of glia and rare neural progenitor cells (Fig. 1A–C) (Fernández-Hernández et al, 2013; Moreno et al, 2015; Simões et al, 2022). To understand whether stab lesions also caused elevated ROS levels, we tested induction of ROS-scavenging glutathione S transferase D1 (GstD1) using a gstD>GFP reporter (Sykiotis and Bohmann, 2008), which is responsive to oxidative stress. While intact brains showed very low *gstD*-driven GFP, levels were strongly upregulated after OL injury (Fig. 1D–H), pointing to elevated ROS. Using a probe that emits fluorescence upon oxidation (CellROX), we also detected local ROS accumulation in injured OLs at acute time points (Fig. 1I–M).

As we previously found that glia act as important coordinators of neuro-glial interactions (Simões et al, 2022), we performed transcriptional profiling (bulk RNAseq) of pools of gfp + glia sorted from injured versus intact OLs of *repo-Gal4; UAS-mCD8::gfp* flies to gain a more comprehensive understanding of glial responses to injury and ROS in particular. Among significantly upregulated genes (Dataset EV1) we found known factors upregulated by brain injury such as the JNK target *mmp1* and several GO terms related to redox-dependent processes (Fig EV1A). We further detected induction of several oxidative stress-related genes (Fig. 1N): (1) glutathione S transferases *gstD1*, *gstO1*, *gstE1*, (2) *g6pd/zw* encoding for the rate-limiting enzyme in the pentose phosphate pathway that produces the critical reducing agent NADPH, and (3) *glaz*, shown to regulate neuro-glial lipid-shuttling and resilience to ROS (Liu et al, 2017; Walker et al, 2006; Sanchez et al, 2006). Together, these findings support that glial cells rapidly activate antioxidant proteins and metabolic adaptations, suggesting that the upregulated components form part of the brain's concerted protective response to mitigate oxidative stress.

Membranes of brain cells contain abundant polyunsaturated fatty acids (PUFAs), which are highly susceptible to lipid peroxidation initiated by ROS. To understand the potential harmful impact of stab lesion-induced ROS, we examined lipid peroxidation with the ratiometric PUFA peroxidation sensor (C11-BODIPY 581/591) and the neutral lipid stain LipidTox to visualize glial lipid droplets (Fig. 1O–U). Non-injured brains (OLs and central brain) contained mainly non-oxidized (non-Ox) lipids (Figs. 1O,P,U and EV1B,C,H). In contrast, brains treated with the ROS-producing agent paraquat showed a high degree of oxidized (Ox) lipids, as expected (Figs. 1Q,R,U and EV1D,E,H). Injured OLs showed intermediate Ox/non-Ox ratios (Fig. 1S–U) compared to lipids in the central brain, not targeted by the lesion, which showed very low peroxidation (EV1F–H). The partial lipid peroxidation in injured OLs suggested that the activated antioxidant responses are efficient in restricting ROS damage as a consequence of small stab lesions. The induced programs likely complement general brain protective mechanisms such as the sequestering away of polyunsaturated acids from membranes into glial lipid droplets, which were shown to shield glia and neural stem cells from ROS during nervous system development (Bailey et al, 2015).

## Injury-associated ROS derive from the membrane NADPH oxidase Duox

To understand the origin of injury-induced oxidative stress, we first evaluated a mitochondrial source as mitochondrial dysfunction and ROS have been linked to brain damage. We examined transgenic lines expressing MitoTimer (Laker et al, 2014), a sensor protein that irreversibly shifts to red when oxidized, thereby providing a record of cumulative redox events within mitochondria. We drove MitoTimer with the OL driver *GMR-Gal4* or induced expression prior to injury with ubiquitous *tubulin-Gal4* in adults with a temperature shift, leading to degradation of the thermosensitive (ts) Gal4 suppressor Gal80[ts]. In both cases, we found only mild (*GMR*) or no changes (*tub-Gal4*) in (oxidized) MitoTimer in response to injury, whereas aging produced significantly higher levels than injury (Figs. 2A–D and EV2A–D). To further evaluate the results, we next expressed an ultrasensitive ratiometric ROS reporter targeted to the mitochondria (*UAS-mito::roGFP2::Tsa2ΔCpΔCr*) (Sobrido-Cameán et al, 2025; Morgan et al, 2016), but did not detect a significant increase in mitochondrial ROS in injured versus control brain areas (Figs. 2E and EV2E,F). These findings suggested that mitochondrial ROS production is not the major driver of oxidative stress upon acute brain injury by stab lesion.

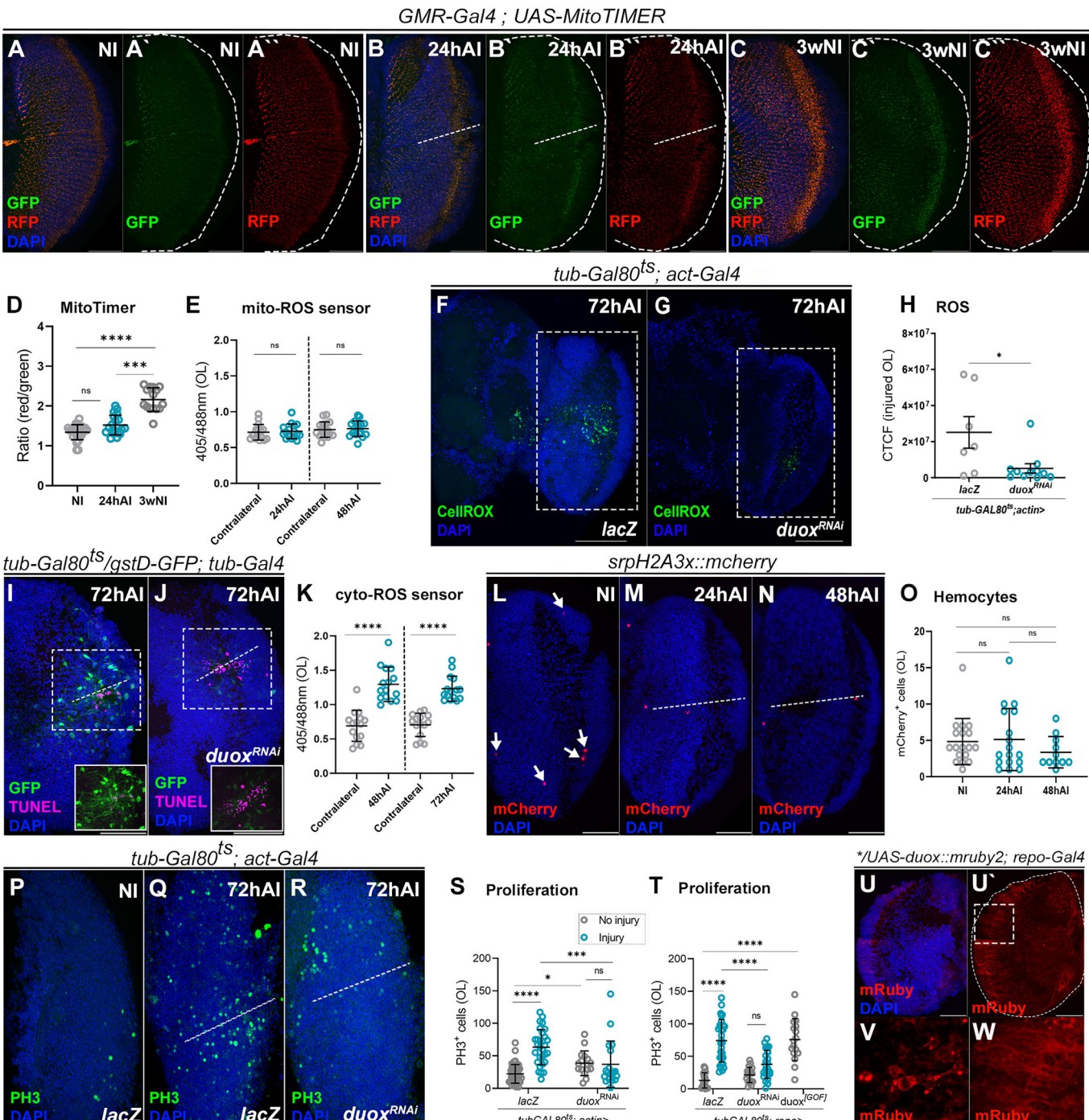

The relatively broad activation of the antioxidant response revealed by *gstD* > GFP (Fig. 1E) was compatible with an extracellular release and diffusion of ROS. We therefore considered NADPH oxidases (Nox, Duox) that use NADPH as substrate and are each encoded by a single gene in the fly. Transcriptional single-cell resources showed that *duox* is prominently expressed in glial subsets in the OLs (NCBI-GEO: GSE142787) (Ferreira and Desplan, 2023), whereas *nox* transcripts are only detected in very few brain cells. We therefore tested whether knock-down of *duox* would prevent the ROS peak after injury. We selected a *duox* RNAi line that was shown to significantly suppress

ROS formation (Kizhedathu et al, 2021), activated it at the adult stage with *tub-Gal80^{ts}; act-Gal4* and found that ROS were significantly reduced after injury compared to control brains (Fig. 2F–H). To corroborate this result, we further knocked-down *duox* prior to injury with the above tools, but in combination with the *gstD-GFP* reporter. *Duox* RNAi strongly reduced oxidative stress compared to control or *nox* RNAi for lesions that caused a comparable amount of apoptosis (TUNEL) (Figs. 2I,J and EV2G).

Duox produces extracellular reactive superoxide ($O_2^{\bullet -}$) from intracellular oxygen and NADPH, which is converted into more

**Figure 2. Glial Duox activity promotes proliferation in the injured brain.**

(A–C) GMR-Gal4-driven MitoTimer in non-injured (NI), injured (24 h AI) and aged OLs (3-week-old flies). OL Images show levels of green (non-oxidized) and red (oxidized) MitoTimer. Nuclei are stained with DAPI (blue). Dashed white lines indicate stab lesions in all images. (D) MitoTimer ratios (red-oxidized/green-non-oxidized). Data is shown as mean ± SD. ****$P < 0.0001$, ***$P < 0.001$ ($P = 0.0002$) and n.s. ($P = 0.51$) by a nonparametric Kruskal–Wallis test with Dunn's multiple comparison correction. $N = 25$ (NI), 19 (24 h), 15 (3w) OLs. (E) Dot plots of mitochondrion-targeted ratiometric mito::roGFP2::Tsa2ΔCpΔCr sensor for injured OL versus control areas (contralateral side). Data are shown as mean ± SD. n.s. ($P = 0.99$) by paired $t$ test. $N = 16$ (24hI), 16 contrl. and $N = 18$ (48hI), 18 (contrl). (F, G) ROS signal (CellRox, green) in OLs with adult-onset *duox-RNAi* with *tub-Gal80[ts]*, *act-Gal4* or controls (*UASlacZ*) 72 h AI. Nuclei are shown in blue. Boxed areas represent the analyzed area (R.O.I). (H) ROS quantification in OLs. CTCF corrected total cell fluorescence. Data are shown as mean ± SD. *$P < 0.05$ ($P = 0.044$) by nonparametric Mann–Whitney $U$ test. $N = 7$ (lacZ), 11 (duox RNAi) OLs. (I, J) Reduced gstD-gfp signal in injured OLs with adult-onset *duox* RNAi. Dying cells at the injury sites (white dashed line) are labeled in magenta (TUNEL). DAPI stains nuclei (blue). (K) Dot plots of *tub>gal4*-driven ratiometric roGFP2::Tsa2ΔCpΔCr sensor activated at adult stage, 48 h and 72 h AI, for injured versus non-injured areas on the contralateral lobe. Data are shown as mean ± SD. ****$P < 0.0001$ by paired $t$ test. $N = 15$ for both samples. (L–N) Hemocytes (*srpH2A3x::mCherry*) (arrows) in intact (L) and lesioned OLs (M, N). DAPI stain nuclei (blue). (O) Hemocyte quantification per OL. Data is shown as mean ± SD. n.s. by nonparametric Kruskal–Wallis test with Dunn's multiple comparison correction. $N = 18$ (NI), 17 (24hAI) and 11 (48hAI) OLs. (P–R) PH3 signal in non-injured (NI) and injured OLs (AI) with conditional *duox* RNAi activation (R) versus control (*UASlacz*) (Q). (S) Conditional activation of *duox*RNAi with *tub-Gal80[ts]*; *act-Gal4* (P). Quantification of proliferation (PH3+ cells/OL) 72 h AI vs control (NI). Data is shown as mean ± SD. Two-way ANOVA on log-transformed data with Tukey's multiple comparison test was applied. ****$P < 0.0001$, ***$P < 0.001$ ($P = 0.00047$), *$P < 0.05$ ($P = 0.013$) and n.s. ($P = 0.6$). $N = 46$ (lacZ), 31 (lacZI), 16 (RNAiduox), 19 (RNAiduoxI). (T) Proliferation (PH3/OL) in non-injured and injured OLs (72 h AI) with adult-onset RNAi or overexpression of *duox* in glia with *tub-Gal80[ts]*; *repo-Gal4*. Data are presented as mean ± SD. Two-way ANOVA on log-transformed data with Tukey's multiple comparison test was applied. ****$P < 0.0001$; n.s. ($P = 0.156$). $N = 17$ (lacZ), 31(lacZI) 20 (RNAiduox), 25 (RNAiduoxI); 18 (OEduox). (U–W) Duox::mRuby (red) induced at the adult stage with **tub-Gal80[ts]*; repo-Gal4*. White boxed area marks inset (W). Nuclei are shown in blue (DAPI). Scale bars are 100 μm for brains, 50 μm for OL images, and 20 μm for insets. Source data are available online for this figure.

stable hydrogen peroxide ($H_2O_2$) that can diffuse in the tissue. To trace the effect of $H_2O_2$, we employed the hydrogen peroxide-specific sensor *UAS-roGFP2::Tsa2ΔCpΔCr* (Sobrido-Cameán et al, 2025) and detected increased ROS levels in brain tissue in the injured OL area compared to control areas on the contralateral side (Figs. 2K and EV2H,I).

Altogether, these experiments identify Duox-derived hydrogen peroxide as the likely causative agent of oxidative stress in response to acute brain damage. These findings also support the view that sources and temporal dynamics of oxidative stress may be markedly different in acute brain injury versus chronic neurodegenerative disease, where mitochondrial ROS are strongly involved.

## Glial $H_2O_2$ acts as a mitogenic signal in the injured fly brain

In fly epithelial wounds, ROS have been found to act as chemoattractive signals for hemocytes (Razzell et al, 2013; Moreira et al, 2010). To understand if ROS could attract immune cells from circulation to brain lesions, we quantified hemocytes with bright *srpH2A3xmCherry* (Gyoergy et al, 2018), marking all hemocytes, but did not detect differences in hemocyte presence in intact versus injured OLs (Fig. 2L–O). This was also the case for *hml*-Gal4-driven GFP, which labels a subset of adult hemocytes (Fig. EV2J–L), indicating that small stab lesions do not cause significant immune cell recruitment.

We therefore decided to study if ROS modulated the regenerative response as $H_2O_2$, able to diffuse across cell membranes or being imported by aquaporin channels (Sobrido-Cameán et al, 2023), can alter signaling in surrounding cells. To this end, we conditionally suppressed Duox function at the adult stage with a ubiquitous driver and assessed injury-induced proliferation by quantifying the signal of the mitotic marker phospho-histone H3 (PH3) present in lesioned optic lobes (Fig. EV2M) (Simões et al, 2022). We found that *duox* RNAi significantly suppressed mitotic counts in injured brains compared to controls (Fig. 2P–S). As *duox* transcripts have been mainly detected in glia in the adult brain, we next activated *duox* RNAi specifically in glial cell types with *tub-Gal80[ts]*; *repo-Gal4*, which strongly reduced cell divisions after injury, albeit not completely (Fig. 2T). As an

increase in ROS can stimulate the proliferation of neural stem cells (Sies et al, 2022; Bigarella et al, 2014), we assessed the effect of *duox* RNAi in Dpn-expressing adult neural progenitors driven by *dpnT2A-Gal4; tubGal80[ts]* (Simões et al, 2022; Li et al, 2020). Duox suppression in progenitors did not significantly reduce injury-induced cell divisions compared to control flies. In turn, non-injured and injured *dpn>duox* RNAi OLs showed similar mitotic counts, suggesting that Duox function may be needed in progenitors to support their proliferation (Fig. EV2N–Q). However, it is likely that ROS released from more abundant glial cells can also considerably stimulate progenitor activation in a cell non-autonomous manner when Duox function is suppressed in progenitors.

To understand if increased ROS release by glia is sufficient to induce proliferation without injury, we overexpressed an active form of *duox* (*duox*[GOF]) (Sobrido-Cameán et al, 2023) in adult glia. This indeed resulted in extra cell divisions, suggesting that glial ROS release can stimulate proliferation in the adult brain (Fig. 2T). To better understand Duox localization in membrane compartments, we expressed a previously described mRuby-tagged Duox form (*duox::mRuby2::HA*)(Sobrido-Cameán et al, 2023) and activated expression in glia at the adult stage. We found that Duox::mRuby was extensively distributed along glial protrusions, suggesting a potentially broad presence of Duox in glial networks covering large areas of the OL (Fig. 2U–W).

Taken together, these findings reveal that Duox activity on glial membranes is required and sufficient to stimulate cell divisions in the adult fly brain.

## Strong ROS suppression affects regeneration

The above experiments based on genetic ROS suppression (*duox* RNAi) pointed to a pro-regenerative role of hydrogen peroxide. To evaluate the effect with an alternative method, we next used biochemical ROS suppression with N-Acetylcysteine (NAC), a ROS scavenger that also replenishes intracellular glutathione levels. Control or NAC-containing food was fed to perma-twin flies, in which mitotic-dependent labeling with *actin*-driven flippase—based on the twin-spot MARCM (Fernández-Hernández et al, 2013)—was activated at the

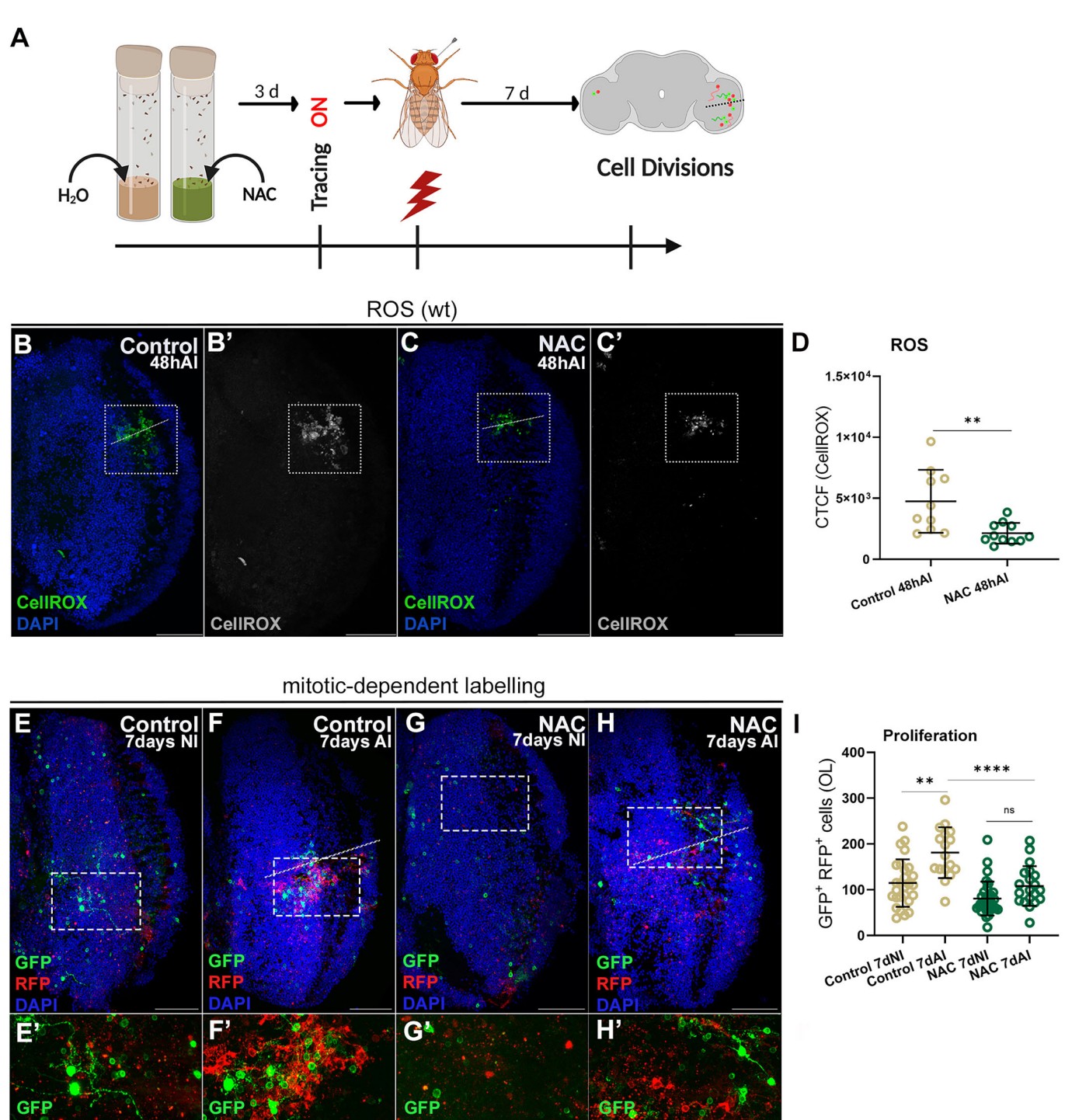

**Figure 3. Duox function and ROS are required for regeneration.**

(A) Scheme depicting the activation of mitotic-dependent cell labeling prior to injury by a temperature shift (18–29 °C) in flies raised on a control or antioxidant NAC diet. (B, C) ROS detection in OLs of flies fed with control (B) or NAC diet (C). Nuclei are counterstained with DAPI (blue). The boxed area represents areas of insets and R.O.I. Dashed white lines indicate stab lesions in all images. (D) ROS Quantification in injured OL (ROI) of NAC-fed and control flies. CTCF, corrected total cell fluorescence. Data are presented as mean ± SD. **$P < 0.01$ ($P = 0.0049$) by parametric unpaired two-tailed Student's t test. $N = 10$ (control) and 11 Ols (NAC). (E–H) Mitotic-dependent labeling (GFP+ or RFP + ) in non-injured OLs (E, G) or lesioned OLs (F, H) 7 days after adult-onset activation in flies fed with control (E, F) or NAC diet (G, H). Genotype of flies: *w; FRT40A, UAS-CD8-GFP, UAS-CD2-Mir/ FRT40A, UAS-CD2-RFP, UAS-GFP-Mir; act-Gal4 UAS-flp/tub-Gal80ts*. Nuclei are marked by DAPI (blue). Insets from the injury site (boxed area) are shown below. (I) Quantification of GFP + /RFP+ marked cells per OL. Data are shown as mean ± SD. ****$P < 0.0001$, **$P<0.01$ ($P = 0.0019$) and n.s. ($P = 0.063$) by Kruskal–Wallis ANOVA with Dunn's correction for multiple comparisons. $N = 17$ (control), 20 (control-I), 32(NAC), 20 (NACI). Scale bars are 50 μm for OL images, and 20 μm for insets. Source data are available online for this figure.

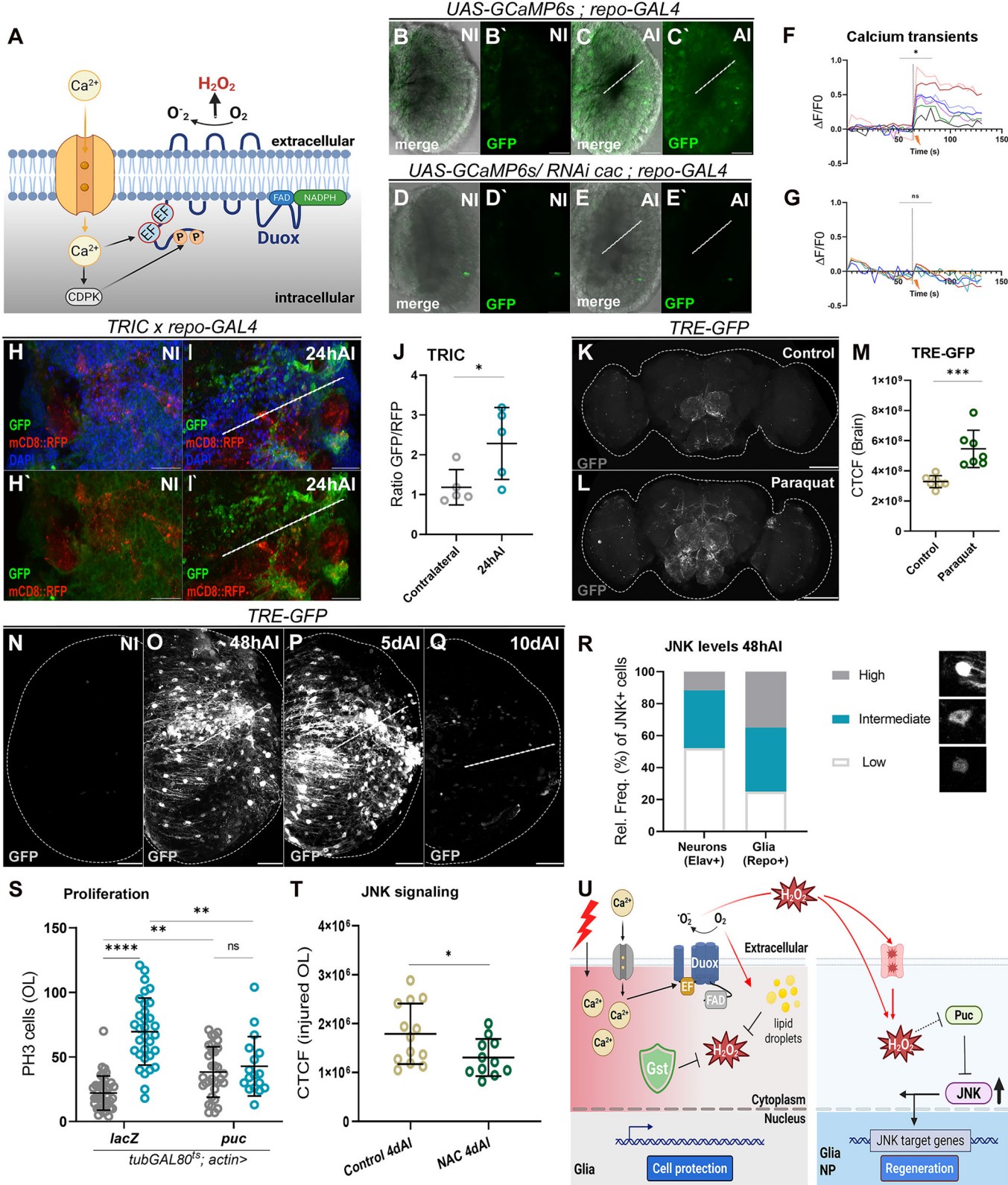

Figure 4.  Calcium and ROS contribute to pro-regenerative JNK activity.

(A) Scheme of the NADPH oxidase Duox with intracellular NADPH oxidase (green) and calcium-modulated EF hand domain (light blue). (B, C) GCaMP signal in OL glia in brain explants prior to injury (B) and after injury (C). Dashed white line marks injury site in all images. (D, E) GCaMP signal of OLs (brain explants) with *cac* RNAi in glia before (D) and after injury (E). (F) Injury-induced calcium transients in glia before and after injury. Data are represented as a fluorescence change over baseline signal. *$P < 0.01$ ($P = 0.0156$) by nonparametric paired Wilcoxon signed-rank test comparing peak values between conditions. $N = 7$ brains. (G) Injury-dependent calcium transients are reduced with *cac* RNAi. Data are represented as a fluorescence change over baseline signal. n.s. ($P = 0.063$) by a nonparametric paired Wilcoxon signed-rank test comparing peak values between conditions. $N = 5$ brains. (H, I) Calcium levels (GFP) in glia near the lesion site (I) compared to the non-injured control area (H) as revealed with *repo-Gal4*-driven TRIC. Membrane RFP signal shows calcium-independent markage (H). Nuclei are counterstained with DAPI (blue). (J) Plot of mean GFP/RFP fluorescence. Data is shown as mean ± SD. *$P < 0.5$ ($P = 0.04$) by parametric paired two-tailed Student's *t* test. $N = 5$ OLs per condition. (K, L) Brain TRE-GFP levels upon paraquat feeding (L) versus control (K). (M) TRE-GFP fluorescence intensity in brains of control and paraquat-fed flies. Data is shown as mean ± SD. ***$P < 0.001$ ($P = 0.0008$) by unpaired two-tailed Student's *t* test. $N = 7$ brains per condition. (N–Q) TRE-GFP activation before (N) and after injury (O–Q). (R) Graph depicting the relative frequency of neuronal and glial cells with high, intermediate and low JNK activity 48 h after injury. Overlay with Elav (neuronal marker) and Repo (glial marker) are shown in EV4H. $N = 16$ OLs. (S) Injury-induced proliferation in flies with conditional overexpression of puckered (puc) vs controls. Data is shown as mean ± SD. Two-way ANOVA on log-transformed data with Tukey's multiple comparison test was applied. ****$P < 0.0001$, **$P < 0.01$ ($P = 0.001$), *$P < 0.05$ ($P = 0.016$) and n.s. ($P = 0.8$). $N = 40$ (*lacZ*), 36 (*lacZI*), 40 (*puc*); 17 (*pucI*). (T) JNK activity 4 days AI on control and NAC diets. Data are presented as mean ± SD. *$P < 0.05$ ($P = 0.038$) by unpaired two-tailed Student's *t* test. $N = 13$ (control), 11 (NAC) OLs. CTCF, corrected total cell fluorescence. (U) Model of Calcium-ROS stimulated regeneration by Duox. Injury triggers calcium elevation and Duox-dependent ROS generation in glia, activating antioxidant factors and sustaining proliferation, promoting JNK activity. Scale bars are 100 μm (B–E), and 50 μm for OL images, and 20 μm for insets. Source data are available online for this figure.

adult stage prior to injury, resulting in labeling of cells that have divided with membrane GFP or RFP (Fig. 3A).

We first fed wild-type flies with NAC-supplemented food (NAC + , 100 μg/ml), which significantly reduced ROS levels in injured OLs compared with flies raised on control food (H₂O + ), confirming effective ROS suppression (Fig. 3B–D). We next quantified the number of fluorescently labeled cells (GFP/RFP) in injured (7 days AI) and intact OLs of flies maintained on either NAC or control diet (Fig. 3E–I). Flies on control food exhibited a significant increase in cell divisions following injury, relative to non-injured brains. In contrast, the injury-induced increase was absent when ROS were quenched with NAC. These results indicate that ROS suppression markedly delays or attenuates the regenerative response.

Feeding of flies with high Vitamin C (250 μg/ml), combined with excess Vitamin E (20 μg/ml) also appeared to reduce injury-dependent cell divisions (Fig. EV3A–C), but the effect was less pronounced compared to NAC and baseline signal in non-injured OLs was not recorded in this case. It is noteworthy that non-injured brains showed considerable labeling. The signal can reflect homeostatic proliferation, but the absence of an obvious twin cell in some cases suggests that there may be also some background, potentially due to non-reliable knock-down of GFP/RFP by the microRNAs in the twin-spot MARCM. In the future, it would be desirable to have a new version of a mitotic-dependent tracing tool, where the marking of cells is not dependent on the twin spot cassettes.

We also tested whether overexpression of a single ROS-scavenging enzyme (Catalase or Sod2) would impact proliferation as measured by PH3 staining (Fig. EV3D), but this was not the case. This may be due to the fact that efficient ROS suppression may require simultaneous overexpression of both components, as observed in wing discs (Santabárbara-Ruiz et al, 2015).

Taken together with the previous findings on Duox inhibition, the results show that ROS act as a pro-mitotic signal, as dampening of the injury-induced ROS peak blunts the regenerative response.

## ROS sustains pro-regenerative JNK activity

The calcium-sensitive intracellular domain of Duox has been shown to be essential for ROS-generation, acting as a driving force of Duox activity and ROS bursts (Razzell et al, 2013) (Fig. 4A). We therefore asked if acute brain injury provoked a calcium increase, especially in Duox-expressing glia. By preparing fresh brain explants, we measured increased intracellular calcium in OL glia shortly after injury with the genetically encoded calcium sensor GCaMP6 (Chen et al, 2013) and detected elevated calcium in glia after injury (Fig. 4B,C,F). As a candidate component that could be involved in calcium responses, we examined the role of Cacophony forming the primary subunit of a voltage-gated calcium channel that, apart from neurons, is also expressed in *duox*-expressing glia (Ferreira and Desplan, 2023). RNAi of *cac* attenuated the calcium signal in glia (Fig. 4D,E,G), suggesting that membrane calcium channels are involved in mediating calcium entry into glia upon injury. We also examined injury sites in the OL in flies expressing the transcriptional reporter of intracellular calcium (TRIC) (Gao et al, 2015) and found glia with elevated calcium signal in the majority of lesions (Fig. 4H–J). Thus, we conclude that injury-triggered calcium transients can support Duox activation in injury-exposed glia.

ROS released from glia upon injury have the potential to modulate pro-regenerative signaling. As a potential pathway, we examined JNK signaling and first tested if ROS increase, without mechanical injury, can trigger increased activity. Flies were fed with either 5% sucrose or 15 mM paraquat in 5% sucrose. As expected, paraquat caused elevated JNK stress signaling in the gut as measured by the sensitive *TRE::gfp* reporter, which contains several AP-1 binding sites fused to *gfp* (Chatterjee and Bohmann, 2012) (Fig. EV4A–C). However, we also detected increased JNK levels in the adult brain after paraquat feeding, together with higher levels of oxidative stress (Figs. 4K–M and EV4D–F). We thus monitored the temporal pattern of JNK activity in the brain in more detail and remarkably found broad and perduring JNK activity even 5 days after injury, both in neurons (Elav + ) and glia (Repo + ) with levels returning to baseline only around 10 days after injury (Figs. 4N–Q and EV4G,H). Numerous cells displayed intermediate or low JNK activity, compatible with driving regeneration (Pinal et al, 2019) (Fig. 4R). To test more directly if JNK signaling regulates regeneration, we measured injury-dependent proliferation in brains with induced overexpression of the JNK phosphatase

Puckered (Puc), which negatively regulates JNK (Martín-Blanco et al, 1998). This led to a significant reduction of cell divisions (Fig. 4S), showing that JNK activity is important to promote regeneration.

ROS can increase kinase activity, including JNK, by oxidizing cysteine residues on phosphatases, that restrict kinase activity or on other proteins that keep kinases inactive in non-stressed cells (Averill-Bates, 2024; Lennicke and Cochemé, 2021). We therefore tested if ROS suppression had an impact on injury-triggered JNK activity. We found that JNK pathway activity was similar shortly after brain injury, but lower several days post injury in flies kept on NAC food (Fig. 4T), suggesting that ROS can sustain JNK activity in cells near the injury site. The pro-regenerative effect of ROS on JNK and other pro-regenerative factors has also been observed in regenerating wing epithelial discs (Pinal et al, 2019; Hariharan and Serras, 2017).

In conclusion, these results show that Duox-derived ROS unexpectedly act as positive regulators of the regenerative response, capable of sustaining pro-regenerative JNK activity and required to drive proliferation. Hydrogen peroxide ($H_2O_2$), which results from Duox activity, is a more long-lived ROS and its lack of charge enables diffusion and crossing of membranes (Sies et al, 2022). Based on our results, we propose a model whereby $H_2O_2$ release from glial Duox enhances pro-regenerative JNK levels in nearby glia and neural progenitors, supporting their proliferation (Fig. 4U). Upstream of ROS, injury-induced calcium influx in glia likely acts as the earliest injury signal, which can promote Duox activity while different cellular antioxidant adaptations protect glia against lipid peroxidation. Assessing Duox function at the organismal level is complex because its transient, beneficial activity after injury may be confounded by pro-inflammatory effects during aging. In line with this, we observed that ubiquitous knockdown of *duox* extended lifespan, similar to *duox* heterozygosity (Baek et al, 2022), but had little effect on survival after brain injury (Fig. EV4I). Further work involving *duox* manipulation in glia, combined with brain function assays will be required to better understand how Duox-dependent plasticity influences brain repair and behavior.

The elevated levels of different ROS after acute brain injury in patients represent a major concern due to the oxidative damage caused to lipids and proteins. Nevertheless, suppression of ROS with antioxidant agents has not shown efficacy in ameliorating recovery from brain injury so far (Hall et al, 2019). Our findings support that, besides the damaging effect of free radicals, certain ROS play an underestimated role in activating brain repair responses. Hence, more targeted interventions to mitigate ROS levels may have to be developed to improve brain injury outcomes in the future.

# Methods

### Reagents and tools table

| Reagent/resource | Reference or source | Identifier or catalog number |
| --- | --- | --- |
| **Experimental models** | | |
| wild-type (Canton S) | gift Moreno E | N/A |
| tubGal80ts/Cyo; act-Gal4/TM6b | Rhiner Lab | N/A |

| Reagent/resource | Reference or source | Identifier or catalog number |
| --- | --- | --- |
| tubGal80ts; repo-Gal4/TM2 | Rhiner Lab | N/A |
| tubGal80ts; tubGal4/TM6b | Rhiner Lab | N/A |
| UAS lacZ | gift Moreno E | N/A |
| TRE-GFP-16 (AP1 reporter) | gift Moreno E | N/A |
| gstD-GFP/(CyO); MKRS/TM6B | gift Banerjee U | N/A |
| w; FRT40A, UAS-CD2-RFP, UAS-GFP-Mir; tub-Gal80ts/TM6 | Rhiner Lab | N/A |
| w; FRT40A, UAS-CD8-GFP, UAS-CD2-Mir; act-Gal4, UAS-flp/TM6B | Rhiner Lab | N/A |
| Sp/CyO; srpHemo-H2A::3xmCherry | gift Siekhaus D | N/A |
| hml-Gal4, UAS-GFP | Bloomington center | #6397 |
| nsyb-Gal4/(TM6b) | gift Moreno E | N/A |
| UAS-MitoTimer | Bloomington center | #57323 |
| UAS-RNAi duox ChIII | Bloomington center used for most assays | #32903 |
| UAS-RNAi duox ChII | Bloomington center used for life span | #38907 |
| gstD-GFP/(CyO); RNAi duox/ TM6B | This study | N/A |
| GMR-Gal4 | Bloomington center | #8605 |
| 20×UAS-GCaMP6s | Gift Ribeiro C | N/A |
| TRIC | Bloomington center | #62828 |
| UAS-Catalase | Bloomington center | #24621 |
| UAS-Sod2 | Bloomington center | #24494 |
| if/CyO; repo-GAL4/Tm6b | gift Moreno E | N/A |
| UAS puc 14.E/CyO; MKRS/TM6B | gift Moreno E | N/A |
| duox[GOF] | Gifts Landgraf M | N/A |
| 10xUAS-IVS-dDuox::mruby2::4xHA | Gifts Landgraf M | N/A |
| 10xUAS-IVS-mito::roGFP2::Tsa2ΔCPΔCR | gifts Landgraf M | N/A |
| 10xUAS-IVS-roGFP2:: Tsa2ΔCPΔCR | gifts Landgraf M | N/A |
| UAS RNAi cac | VDRC | 104168 |
| UAS RNAi luciferase | Bloomington center | #31603 |
| UAS RNAi nox | Bloomington center | #32902 |
| gstD-GFP/CyO; RNAi nox/ TM6B | this study | N/A |
| dpnT2AGal4; tubGal80ts | Rhiner Lab | N/A |
| UAS-mCD8::GFP; repo-Gal4 | RNAseq | N/A |
| **Antibodies** | | |
| Phospho-Histone H3 (Ser10) (D7N8E) XP® rabbit mAb #53348 | Cell signaling | (D7N8E) |
| anti-Repo mouse | DSHB | RRID: AB_528448 |
| anti-Elav rat | DSHB | RRID: AB_528218 |
| TUNEL kit - Terminal Transferase | Roche | 11093070910 |
| DAPI | Thermo Scientific | N/A |
| Alexa Fluor 488 rabbit | Invitrogen | RRID: AB_2576217 |
| Alexa Fluor 555 rat | Invitrogen | RRID: AB_141733 |

| Reagent/resource | Reference or source | Identifier or catalog number |
|---|---|---|
| Alexa Fluor 647 mouse | Invitrogen | RRID: AB_2535805 |
| CellROX Green Reagent | Life Technologies | #C10444 |
| **Chemicals, enzymes, and other reagents** | | |
| PBS | Fisher Scientific | 21-040-CV |
| Paraformaldehyde | Thermo Scientific | #28906 |
| Triton-100 | Sigma-Aldrich | X100 |
| Schneider's medium | Biowest | #L0207 |
| Vectashield medium with DAPI | Vector Laboratories | H-1200 |
| Vectashield medium | Vector Laboratories | H-1000 |
| Fetal Bovine Serum | Corning | 35-079-cv |
| LipidTox Deep Red (647 nm) | Molecular Probes | #H34477 |
| C11-BODIPY 582/591 | Invitrogen | #D3861 |
| Paraquat- methyl viologen | Sigma-Aldrich | #856177 |
| Glycerol | Sigma | G5516-500ML |
| Sucrose | Gift from Ribeiro C | N/A |
| Filter paper circles | Whatman | 10311612 |
| Vitamin C (250 µg/ml) | Sigma-Aldrich | A/8882/48 |
| Trolox (an analog of vitamin E; 20 µg/ml) | Sigma-Aldrich | 218940010 |
| N-acetylcysteine (NAC) (100 µg/ml) | Sigma-Aldrich | A7250-25G |
| Water (RNA free) | Fisher BioReagents | BP561-1 |
| RNAse Cleaner | Nzytech | MB16001 |
| Papain | Worthington Biochemical Corporation | #LK003176 |
| 1% UltraPure TM low melting point agarose | ThermoFisher | #16520100 |
| **Deposited data** | | |
| Glial bulk RNAseq | NIH | GEO: GSE309567 |
| **Software** | | |
| Fiji/ ImageJ | | https://imagej.nih.gov/ij/ |
| Zen Digital Imaging for Light Microscopy | ZEISS | RRID:SCR_013672 |
| GraphPad Prism 8 | GraphPad Software | http://www.graphpad.com/ |
| Biorender | | |
| Photoshop | | |
| Nextera | | |
| STAR 2.7.0a. | | |
| RStudio (v1.4.1106; "Tiger Daylily" for Windows) | | |
| R package DESeq2 (v1.31.16) | | |
| **Other** | | |
| filaments (0,1 mm or 0,2 mm) | Fine Science Tools | |
| 880 Zeiss LSM Confocal Microscope | ZEISS | |

| Reagent/resource | Reference or source | Identifier or catalog number |
|---|---|---|
| Agilent 2100 Bioanalyzer | | |
| Nunc Glass Bottom Dish | ThermoFisher | #150680 |

## Materials availability

Generated fly strains can be requested from the corresponding author. In particular cases, sharing of a stock used here will need the consent of the lab that provided it.

## Fly lines and maintenance

Flies (*Drosophila melanogaster*) were raised on standard cornmeal agar medium and maintained in an incubator set at 25 °C, 60% humidity with a 12 h light/12 h dark cycle or at 18 °C when combined with *tubGal80$^{ts}$*, and only 3-day-old adult flies were transferred to 29 °C.

The *Drosophila* strains used were the following, # refer to strains obtained from Bloomington: wild-type (Canton S), *tubGal80$^{ts}$/Cyo; act-Gal4/TM6b, tubGal80$^{ts}$; repo-Gal4/TM2, tubGal80$^{ts}$; tubGal4/TM6b, UAS-mCD8::GFP; repo-Gal4, UAS lacZ, TRE-GFP-16* (AP1 reporter), *gstD-GFP/(CyO); MKRS/TM6B* (gift Banerjee U), *w; FRT40A, UAS-CD2-RFP, UAS-GFP-Mir; tub-Gal80$^{ts}$/TM6B, w; FRT40A, UAS-CD8-GFP, UAS- CD2-Mir; act-Gal4, UAS-flp/TM6B, Sp/CyO; srpH2A::3xmCherry* (gift Siekhaus D), *hml-Gal4, UAS-GFP* (#6397), *nsyb-Gal4/(TM6b), UAS-MitoTimer* (#57323), *gstD-GFP/(CyO); RNAi duox/TM6B; GMR-Gal4* (#8605), *20×UAS-GCaMP6s* (gift Ribeiro C), *UAS RNAi duox* (#32903 for most assays and #38907), UAS *RNAi luciferase* (#31603); UAS RNAi *cac* (VDRC 104168), UAS RNAi *nox* (#32902), TRIC (#62828), *UAS-Catalase* (#24621), *UAS-Sod2* (#24494), *if/CyO; repo-GAL4/Tm6b, UAS puc14.E/CyO; MKRS/TM6B* (gift Moreno E), *duox*[GOF], *10xUAS-IVS-dDuox::mruby2::4xHA, 10xUAS-IVS-mito::roGFP2::T-sa2ΔCPΔCR* and *10xUAS-IVS-roGFP2::Tsa2ΔCPΔCR* (gifts Landgraf M), *dpnT2A-Gal4; tubGal80$^{ts}$*.

## Stab lesions

A thin sterile filament (0,1 mm or 0,2 mm, Fine Science Tools), wiped with 70% ethanol was introduced through the right eye of at least 3–4 days-old adult $CO_2$- anesthetized flies to the level of the medulla in the optic lobe as previously described (Fernández-Hernández et al, 2013).

## Immunohistochemistry and image analysis

Fly brains were dissected in chilled PBS, fixed in 4% PFA for 20 min at RT, washed with PBS-Triton-X 0.4% and incubated with primary (overnight at 4 °C) and secondary antibodies (overnight at 4 °C), or washed with PBS 1×. Adult brains were mounted with a spacer to avoid compression of the tissue. The primary antibodies used were Phospho-Histone H3 (Ser10) (D7N8E) XP® rabbit mAb #53348, Cell Signaling (1:50); anti-Repo (ms) (1:20) and anti-Elav (ms) (1:50) (DSHB). For PH3 staining (Cell Signaling), samples were fixed overnight at 4 °C in 2% PFA, washed and incubated for 48h-

72h with primary Ab. To detect apoptotic cells, brains were dissected 72 h after damage to perform TUNEL labeling (TUNEL kit, Roche). Samples were imaged on an 880 Zeiss Confocal microscope, and images were analyzed using ImageJ and Zen Digital Imaging for Light Microscopy.

## ROS detection and lipid peroxidation

For ROS detection in adult brains, we used CellROX Green Reagent (Life Technologies, #C10444). Adult brains were dissected in Schneider's medium (Biowest, #L0207) at several timepoints after optic lobe injury and incubated for 30 min in medium containing 5 μM CellROX Green Reagent, followed by three washes. Samples were protected from light throughout. Then, they were fixed, followed by three washes in PBS and mounted using Vectashield medium with DAPI (Vector Laboratories).

All experiments for ROS detection and lipid measurements were done with unfixed samples. To measure lipid peroxidation, 5-day-old flies were injured (OL lesion) or left non-injured. Brains were dissected 24 h after injury in chilled Schneider's medium, followed by incubation for 30 min in Schneider's medium with 10% FBS and LipidTox Deep Red (647 nm) (1:200) (Molecular Probes, #H34477), plus 2 μM of C11-BODIPY 582/591 (Invitrogen, #D3861). After 3 washes in Schneider's medium, the samples were mounted in the same medium for confocal microscopy analysis. For Paraquat treatments, the brains were incubated for 2 h in 20 mM Paraquat/Schneider's solution, followed by 3 washes in Schneider's medium, and the protocol continued as described above. CNS Lipid peroxidation was quantified by calculating the ratio between the green channel (488/510 nm, oxidative state) and the red channel (581/591, non-oxidized state). Lipids were detected by LipidTox. Ratio maps were generated with MATLAB software plugins. For statistical analyses, LipidTox-positive droplets were selected and quantified with a macro in ImageJ and droplet ratios calculated in MATLAB.

## ROS sensors

For Imaging of the ratiometric ROS sensors, *UAS-mito-roGFP2::T-sa2ΔCPΔCR* and *UAS-roGFP2::Tsa2ΔCPΔCR* samples were prepared as described in Sobrido-Cameán et al, 2025, adapted to OLs. Instead of PBS, we used Schneider's medium, and OL were fixed for 8 min in 4% paraformaldehyde. Specimens were imaged the same day on a Zeiss 880 LSM confocal. The ROS reporter was excited sequentially at 405 and 488 nm with emission detected at 500–600 nm and Z-stack images maximally projected. For each optic lobe, three identical regions of interest (ROIs) were selected within the injury site. In the contralateral optic lobe, ROIs were placed in anatomically matched areas avoiding areas with interfering trachea. Background fluorescence correction was applied prior to calculating the 405/488 ratio.

## Paraquat treatment

In all, 5–7-day-old flies were distributed per vial and fed with 5% sucrose/PBS solution (control) or 15 mM paraquat/PBS (methyl viologen, Sigma-Aldrich, #856177) provided on a filter paper in vials devoid of fly food. Flies were maintained at 25 °C for 24 h or 48 h until dissection of adult brains or guts.

## Antioxidant diets

To prevent ROS production, antioxidants were supplemented into the standard fly food. As antioxidants, we used vitamin C (250 μg/ml), Trolox (an analog of vitamin E; 20 μg/ml), and N-acetylcysteine (NAC) (100 μg/ml), all from Sigma-Aldrich. For the NAC experiments, the solution was added on top of the food, whereas for the vitamins, the solution was added to the fly food after the food had cooled down.

To assess mitotic-dependent labeling, flies of genotype *w; FRT40A, UAS-CD8-GFP, UAS-CD2-Mir/ FRT40A, UAS-CD2-RFP, UAS-GFP-Mir; act-Gal4 UAS-flp/tub-Gal80ts* were raised at 18 °C and 2–3-day-old adult flies were collected and transferred to vials containing control (water) or antioxidant (NAC or vitamins) food for 3 days. Then, they were switched 12 h to 29 °C to activate the tracing system, injured and placed into new vials with the corresponding food. After injury, flies were kept at 29 °C and changed to new vials every 2 days.

## Pooled RNAseq

In total, 400 GFP+ glial cells were harvested from OLs of 5 intact and 5 injured (48hAI) adult brains from *UAS-mCD8::GFP; repo-Gal4* flies. Samples from two high-quality biological replicates were analyzed per condition. OLs were dissected and dissociated into a single-cell suspension as described previously (Simões et al, 2022). For OL dissociation, OLs were incubated with 1 μl/OL of activated Papain (Worthington Biochemical Corporation #LK003176) for 25 min at 25 °C. Total RNA was extracted and processed with the SMART-SEQ2 protocol. RNA integrity was confirmed on an Agilent 2100 Bioanalyzer. Only RNA samples with A260/A280 $\geq$ 2.0, RIN $\geq$ 7.0 and undetectable gDNA levels were used for RNA-seq. Libraries were generated with Nextera and sequenced with NextSeq500 Illumina. Clean reads were mapped to the *Drosophila melanogaster* reference genome (release 6 plus ISO1 MT) using STAR 2.7.0a. All subsequent analyses were done in RStudio (v1.4.1106; "Tiger Daylily" for Windows), based on Ensembl gene IDs. Differentially expressed (DE) genes analysis was accomplished with R package DESeq2 (v1.31.16).

## Calcium imaging of *Drosophila* ex vivo brain

For calcium imaging, flies expressing the calcium sensor 20xUAS-GCaMP6s were crossed to *repo-Gal4* or *20xUASGCaMP6s; repoGal4* flies were crossed to *cac* RNAi flies. Female flies, aged 3–5 days were used for imaging. Flies were anesthetized on ice and dissected in cold Schneider's *Drosophila* Medium (VWR #L00007-500). To expose the brain, the proboscis, antennae, and surrounding cuticle were carefully removed, and the brain rinsed in cold Schneider's *Drosophila* Medium supplemented with 2 mM NaCl before transferring to a Nunc Glass Bottom Dish (ThermoFisher #150680). Brains were kept it place by gently overlaying them with 1% UltraPure™ low melting point agarose (ThermoFisher #16520100) dissolved in Schneider's medium. The agarose was allowed to solidify at room temperature before imaging. Calcium imaging was performed on a Zeiss LSM 880 confocal microscope with a 20x objective with 488 nm excitation. For each brain, a baseline (63 s) was recorded using the time series mode (1 frame/1.26 s) prior to stab lesion with a 0.2 mm filament, followed by

another recording of equal duration after injury. For analysis purposes, the videos were fused back-to-back. Fluorescence intensity was measured in glia using FIJI (ImageJ) from the OL by calculating relative fluorescence change ($\Delta F/F_0$) with $F_0$ corresponding to baseline fluorescence and $\Delta F$ to the fluorescence change relative to $F_0$.

## Quantifications

### *Proliferation*

For mitotic counts based on PH3 staining, RNAi or overexpression was activated 4–5 days before injury by shifting flies to 29 °C, and brains were dissected 72 h after stab lesion. DAPI-positive, PH3 dots were quantified in the adult optic lobes in equal settings across all samples. All PH3 signal was quantified irrespective of signal intensity and signal obtained in non-injured brain considered as baseline (Simões et al, 2022). The controls were performed together.

To assess oxidative stress, we measured CellROX signal (green) using ImageJ. We selected an equal volume of stacks for the injured and non-injured optic lobes. A region distant from the injury site was selected to subtract background fluorescence. To calculate the corrected total cell fluorescence (CTCF), the formula CTCF = Integrated density – (area of selected cells × mean fluorescence of background readings) was applied to projected stacks.

Mitochondrial redox state was evaluated using the MitoTimer reporter by quantifying fluorescence in the green (488 nm) and red (561 nm) channels for intact, injured, and aged (3-week-old flies) using FIJI (ImageJ). Signal quantification was performed using the corrected total fluorescence (CTCF) method, which accounts for background subtraction. The final red/green ratio was calculated from CTCF-corrected values for each optic lobe.

For counts of regenerated cells, all GFP and RFP-positive cells were quantified in equal volumes of injured and non-injured optic lobes in the central OL area. Seven days after Flippase induction in *w; FRT40A, UAS-CD8-GFP, UAS-CD2-Mir/ FRT40A, UAS-CD2-RFP, UAS-GFP-Mir; act-Gal4 UAS-flp/tub-Gal80^ts* flies (Fernández-Hernández et al, 2013).

TRIC signal quantification was performed by calculating the fluorescence ratio between the green (GFP) and red (RFP) channels. For each OL, an identical central ROI containing the injury site and anatomically matched ROI on the contralateral side were defined. Fluorescence intensities were measured using FIJI (ImageJ). The red channel (RFP) served as a membrane-bound marker independent of $Ca^{2+}$ activity, allowing normalization of the green channel (GFP) signal to account for $Ca^{2+}$-independent fluorescence changes. Background correction was applied before calculating the GFP/RFP ratio.

For quantification of JNK positive cells, GFP+ cells in TRE-gfp flies at 48 h AI were divided in three categories (high, medium, and low) and assigned to glia (GFP+Repo + ) or neurons (GFP +Elav + ).

## Statistical analyses

Statistical analyses were performed with GraphPad Prism software and Rstudio (version 2024.12.0). Statistical tests were chosen based on data distribution and homoscedasticity, with parametric or nonparametric approaches applied as appropriate.

## Data availability

The bulk RNAseq data from sorted glia has been deposited in NIH GEO: GSE309567. The source data of this paper are collected in the following database record: Biostudies S-BSST2243.

The source data of this paper are collected in the following database record: biostudies:S-SCDT-10_1038-S44319-026-00703-w.

## Peer review information

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

## Acknowledgements

We thank Matthias Landgraf for sharing ROS sensors prior to publication and David Bailey for construct cloning, the ABBE platform for excellent imaging support, Júlia S for help with JNK reporter analyses, Jingtao Lilue for help with bioinformatics and members of the Rhiner lab for discussion. CA was supported by *HR23-00860* funded by LaCaixa and FCT. This research has been supported by FCT grant FCT-PTDC/BIA-MOL/31170/2017, HR23-00860 from LaCaixa & FCT, the FCT ERC-Portugal program, and the Champalimaud Foundation (113) and facility funding by CONGENTO LISBOA-01-0145-FEDER-022170.

## Author contributions

**Carolina S Alves**: Formal analysis; Validation; Investigation; Visualization; Methodology; Writing—review and editing. **Anabel R Simões**: Formal analysis; Investigation; Visualization; Methodology. **Beatriz Gil Ferreira**: Formal analysis; Investigation; Methodology. **Marta Neto**: Formal analysis; Investigation; Methodology. **Carmo C Soares**: Investigation; Methodology. **Andreia Augusto**: Methodology. **Christa Rhiner**: Conceptualization; Formal analysis; Supervision; Funding acquisition; Writing—original draft; Writing—review and editing.

Source data underlying figure panels in this paper may have individual authorship assigned. Where available, figure panel/source data authorship is listed in the following database record: biostudies:S-SCDT-10_1038-S44319-026-00703-w.

## Disclosure and competing interests statement

The authors declare no competing interests.

# Expanded View Figures

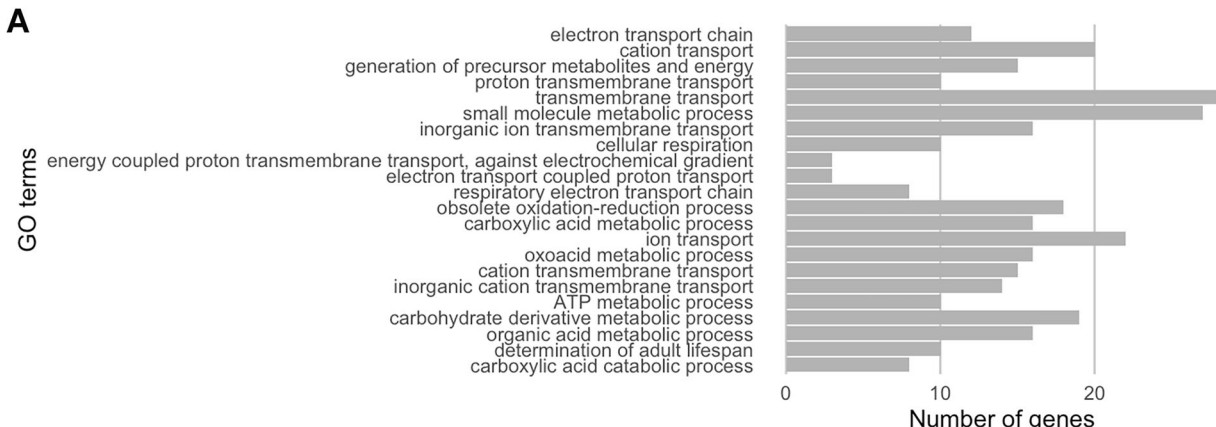

**A**

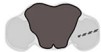

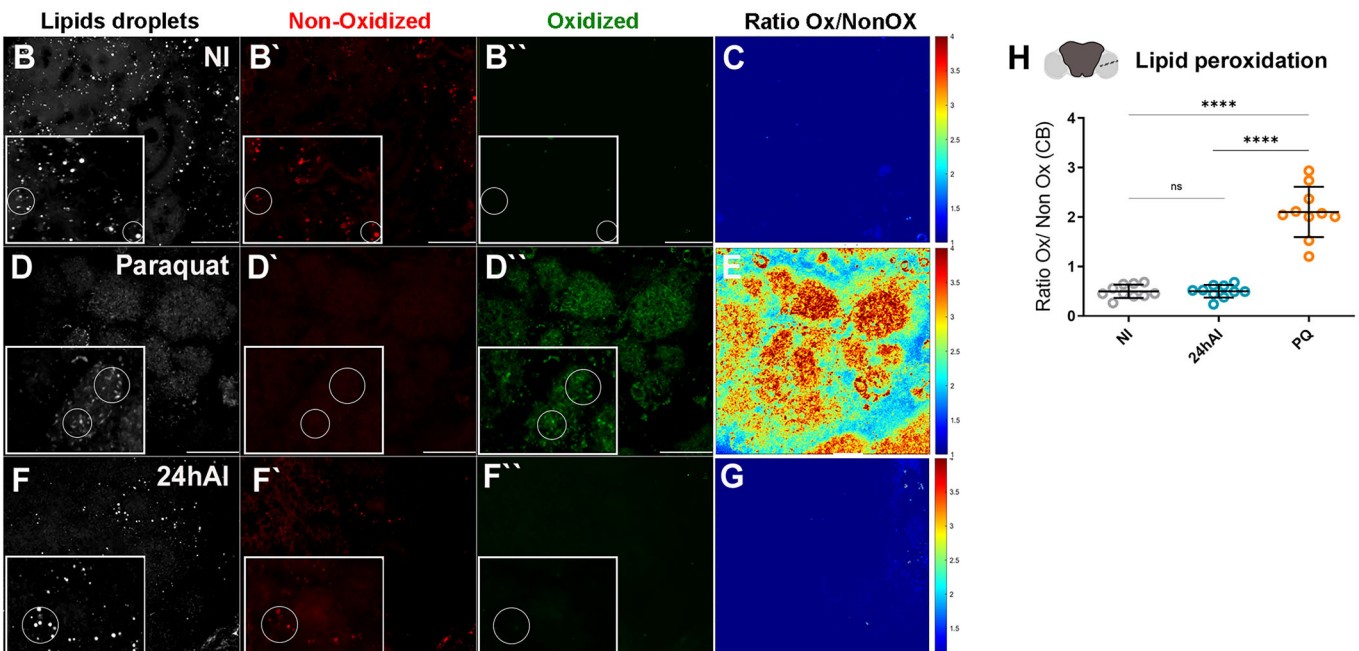

**Figure EV1. Features of injury-exposed glia and lipid droplets.**

(**A**) Enriched Gene Ontology terms for biological processes. (**B, D, F**) Images depict lipid peroxidation in the central brain with the peroxidation sensor C11-BODIPY 581/591 and LipidTox 633 in non-injured (NI) (**B**), paraquat-treated (**D**) and brains with OL injury (central brain unaffected) (**F**). Boxed: Insets depict lipid droplets (white) and non-oxidized (red) and oxidized (green) lipids. Scale bars are 20 μm. (**C, E, G**) Central brain ratios for oxidized/non-oxidized lipids in non-injured brains (NI) (**B**) or brains with OL injury that spares central brain (**F**) and in paraquat-treated control brains (**H**). Lipid droplet peroxidation in the central brain (ratios Ox/non-Ox). Data is shown as mean ± SD. ****$P < 0.0001$ and n.s. ($P = 0.99$) by one-way ANOVA, with Tukey's multiple comparisons correction. $N = 10$ for all conditions. NI non-injured, AI after injury, PQ paraquat-treated.

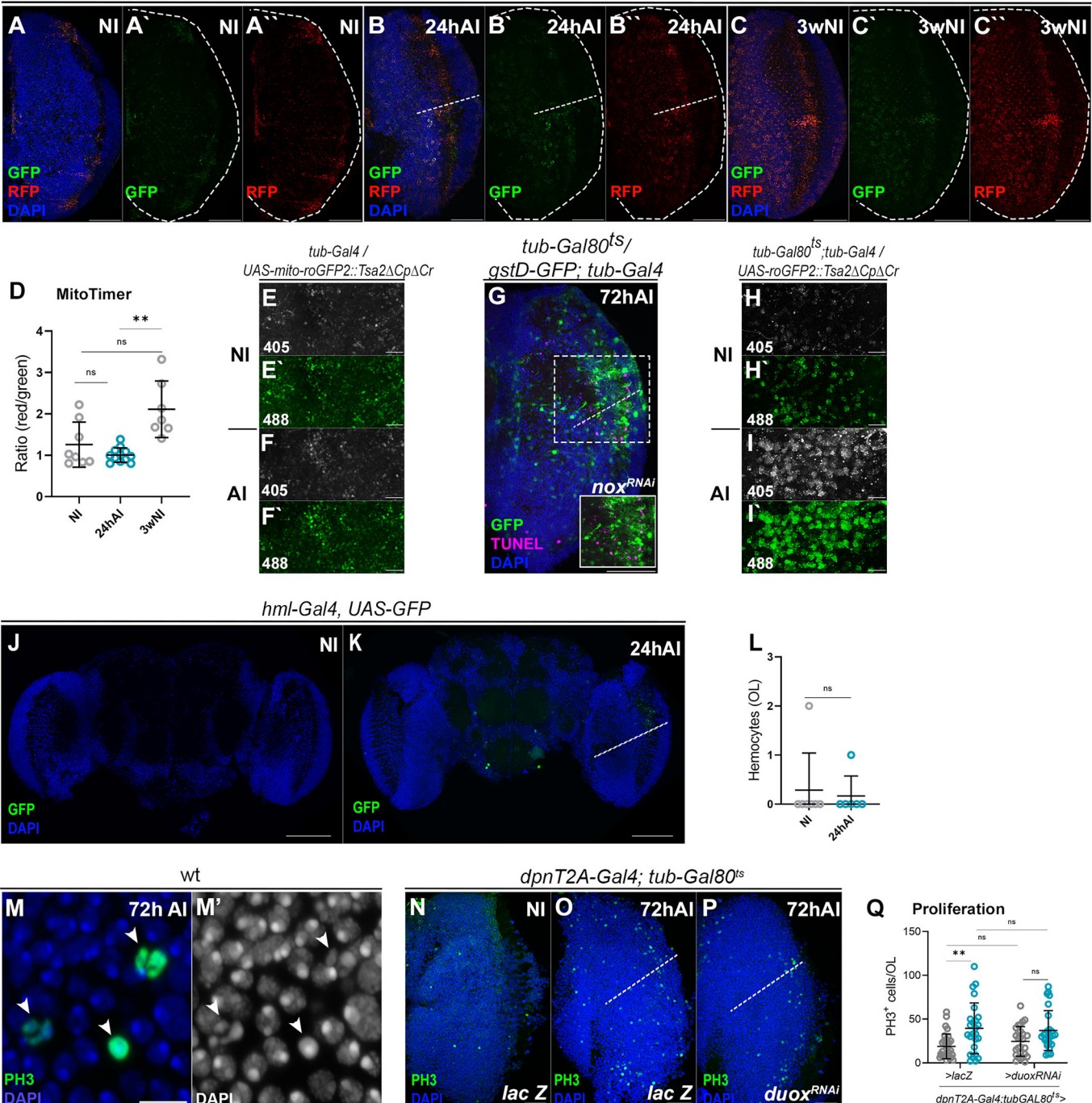

**Figure EV2. ROS sources and function in the injured brain.**

(A–C) *Tubulin*-driven MitoTimer expression in OLs of non-injured, injured (24 h AI) and aged (3weeks-old) flies activated at adult stage. OL Images show levels of green (non-oxidized) and red (oxidized) MitoTimer, Nuclei are shown in blue (DAPI). Dashed white lines mark injury sites in all images. (D) Plots for mitochondrial ROS accumulation. Data are shown as mean ± SD. **P<0.01 (P = 0.0022) and n.s (P = 0.05 NI-3wNI; P = 0.99 NI-24hAI). by Kruskal–Wallis ANOVA and Dunn's correction for multiple comparisons. $N = 8$ (NI), 7(AI), 12 (3w) OLs. (E, F) Images of regions of interest in the non-injured (NI) and injured OLs (48 h AI) with *tub*Gal4-driven mitochondria-targeted *UAS-mito-roGFP2::Tsa2ΔCPΔCR* sensor, 405 nm and 488 nm images. (G) Induction of gstD-GFP at the lesion site in flies with conditional activation of *nox* RNAi. Apoptotic cells are revealed with TUNEL (magenta), nuclei are stained with DAPI (blue). Boxed area marks inset shown below. (H, I) Elevated ROS levels detected with the ratiometric hydrogen peroxide-specific ROS sensor roGFP2::Tsa2ΔCPΔCR in regions of interest in non-injured (NI) and injured OLs (72 h AI) induced at adult stage, 405 nm and 488 nm images. (J, K) Hemocytes visualized by *hml*-Gal4>gfp in non-injured (J) and injured brains (K). (L) Hemocyte quantification (*hmlGal4; UASgfp*) in non-injured and injured OLs (24 h AI). Data n.s. (P = 0.99) by nonparametric Mann–Whitney *U* test. $N = 7$ (NI), 6(AI) OLs. (M) Close-up view of PH3+ cells (green, arrowheads) in control OL (wild-type), cell nuclei are shown in blue (DAPI) (M) or white (M'). (N–P) PH3 signal in OLs in which *duox* RNAi was activated prior to injury in adult neural progenitors with *dpn-T2A-Gal4; tubGal80*[ts] (P) versus activation of *UASlacZ* (control) in intact (N) or injured OLs (O). DAPI stains nuclei (blue). (Q) Proliferation in OLs in which duox was knocked down at adult stage with *dpn-T2A-Gal4; tubGal80*[ts]. Data is shown as mean ± SD. Two-way ANOVA with Tukey's multiple comparison test was applied. **P < 0.01 (P = 0.0022) and n.s. (P = 0.148 *duoxRNAi-duoxRNAil*; P = 0.73 *lacZ-duoxRNAi*, P = 0.97 *lacZI-duoxRNAil*). $N = 34$ (*lacZ*), 16 (*lacZI*), 30 (*duoxRNAi*); 28 (*duoxRNAil*). Scale bars are 100 µm for entire brains, 50 µm for OL images, 20 µm for OL insets and 10 µm for insets of PH3+ cells.

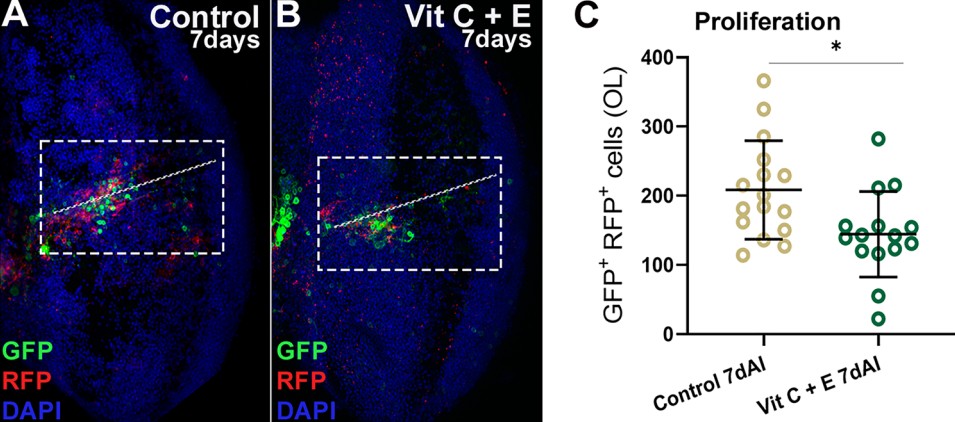

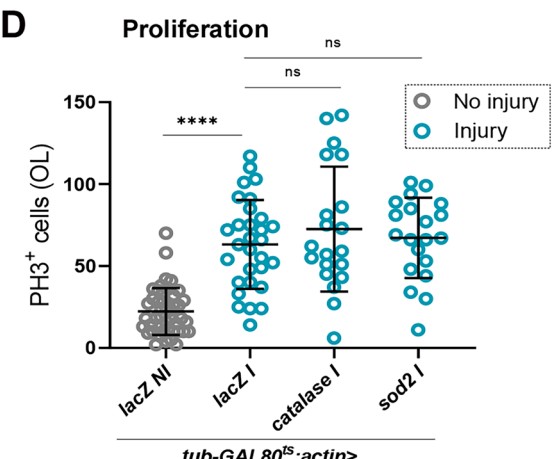

**Figure EV3.  Potential modulatory role of anti-oxidant vitamin diet on regeneration**

(**A**, **B**) Mitotic-dependent labeling (GFP + /RFP + ) in lesioned OLs on control (**A**) or high VitC+E-containing food (**B**). Nuclei are marked by DAPI (blue). White dashed lines represent injury and boxed area the R.O.I. used for analyses. (**C**) Quantification of mitotic-dependent labeling. Data is presented as mean ± SD. *$P < 0.5$ ($P = 0.0123$) by nonparametric Mann–Whitney $U$ test. $N = 16$ (control), 15 (VitC+E) OLs. (**D**) Quantification of PH3 signal per OL in injured control brains (*UASlacz*) or brains where *catalase* or *sod2* were knocked-down in adult flies prior to injury. Data is presented as mean ± SD. ****$P < 0.0001$ and n.s ($P = 0.57$ *lacZI-catalaseI*; $P = 0.95$ *lacZI-SodI*). by Kruskal–Wallis test with Dunn's correction for multiple comparisons. $N = 46$ (*lacZ*), 31 (*lacZI*); 21 (*catalase*), 22 (*sod*). Scale bars are 50 µm for OL images.

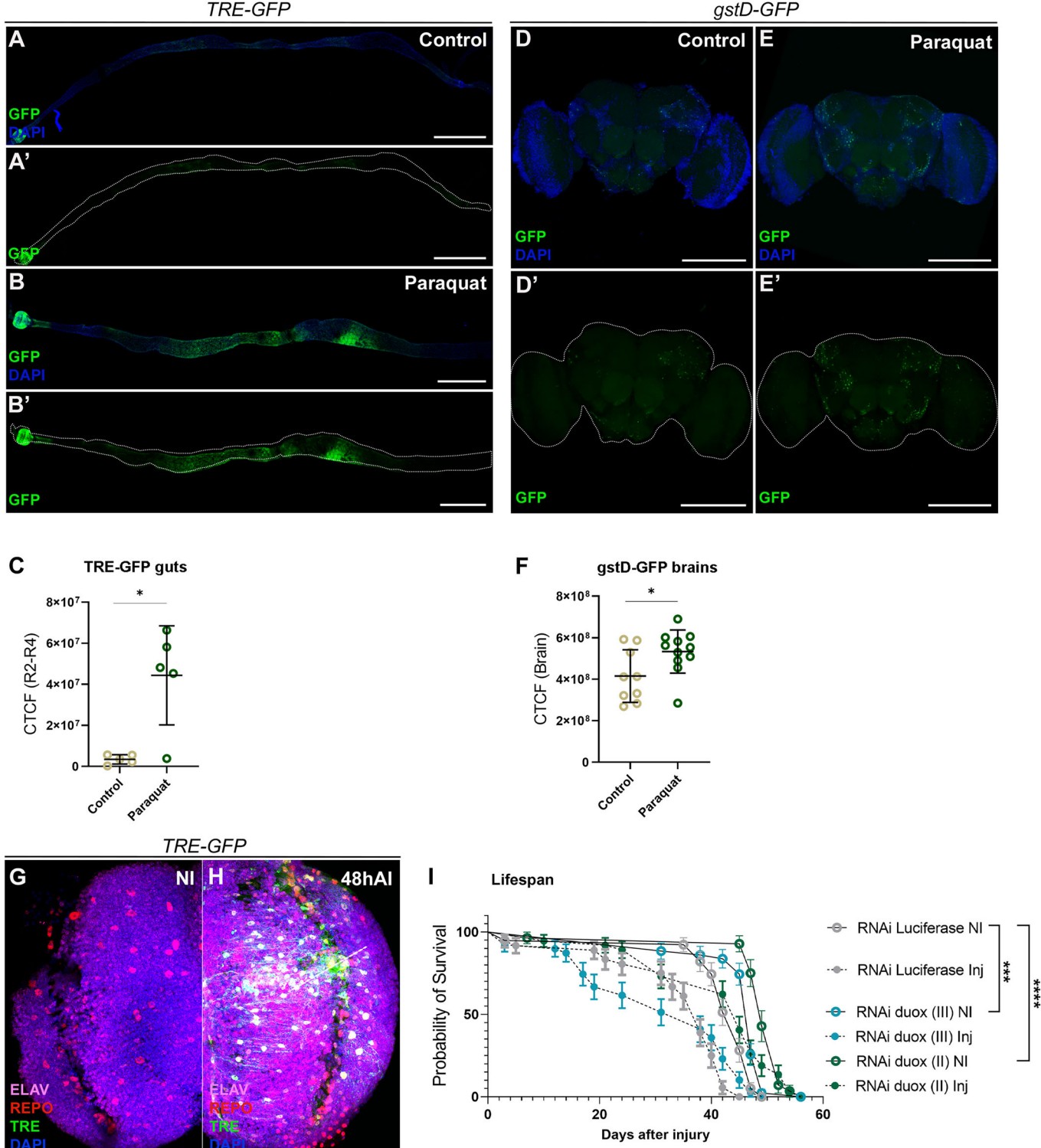

**Figure EV4. Paraquat induces JNK signalling and oxidative stress.**

(A, B) JNK pathway activity (TRE::GFP) in guts of flies fed with control (A) or paraquat-containing food (B). (C) Graph of TRE::GFP signal intensity. Data is presented as mean ± SD. *P < 0.05 (P = 0.032) nonparametric Mann–Whitney U test. N = 6 for both conditions. (D, E) gstD-GFP induction in the brain of paraquat-fed flies (E) versus controls (D). (F) Quantification of gstD-GFP in brains. Data is presented as mean ± SD. *P < 0.05 (P = 0.035) by unpaired two-tailed Student's t test. N = 9 (control), 11 (paraquat) brains. (G, H) TRE-GFP signal was evaluated separately for Repo+ glia (red) and Elav+ neurons (magenta) in non-injured (G) and injured (H) OLs. Dashed white lines indicates stab lesion. (I) Lifespan measurements for *tub>Gal4*-driven control *RNAi (luciferase)* and two different *duox* RNAi lines. Median survival ****P < 0.0001, ***P < 0.001 (***P = 0.0003) by Gehan-Breslow-Wilcoxon test. N = 39 (RNAiluc), 36 (RNAilucI), 43 RNAiduoxIII, 39 (duoxIII-I), 28 (RNAiduoxII), 37 (RNAiduoxII-I). Scale bars are 100 µm and 50 µm for the OL.

