## [Peer Review File · EMBO Reports]

Duox-driven ROS release by glia promotes regeneration in the adult *Drosophila* brain

Christa Rhiner, Carolina Alves, Anabel Simões, Beatriz Ferreira, Marta Neto, Carmo Soares, and Andreia Augusto

Corresponding author(s): Christa Rhiner (christa.rhiner@research.fchampalimaud.org)

Review Timeline:

Submission Date:	2nd May 25
Editorial Decision:	11th Jun 25
Revision Received:	20th Oct 25
Editorial Decision:	4th Dec 25
Revision Received:	18th Dec 25
Accepted:	22nd Jan 26

Editor: Esther Schnapp

Transaction Report:

Dear Christa,

Thank you for the submission of your manuscript to EMBO reports. We have now received the full set of referee reports that is pasted below.

As you will see, the referees acknowledge that the findings are potentially interesting. However, they also have several concerns and suggestions for how the study should be improved and the data strengthened. I think all suggestions are good and should be addressed, and I agree with your proposed revision plan.

I would thus like to invite you to revise your manuscript with the understanding that the referee concerns must be fully addressed and their suggestions taken on board. Please address all referee concerns in a complete point-by-point response. Acceptance of the manuscript will depend on a positive outcome of a second round of review. It is EMBO reports policy to allow a single round of major revision only and acceptance or rejection of the manuscript will therefore depend on the completeness of your responses included in the next, final version of the manuscript.

We realize that it is difficult to revise to a specific deadline. In the interest of protecting the conceptual advance provided by the work, we recommend a revision within 3 months (11th Sep 2025). Please discuss the revision progress ahead of this time with the editor if you require more time to complete the revisions.

- 1) A data availability section providing access to data deposited in public databases is missing. If you have not deposited any data, please add a sentence to the data availability section that explains that.
- 2) Your manuscript contains statistics and error bars based on $n=2$. Please use scatter blots in these cases. No statistics should be calculated if $n=2$.

3) We replaced Supplementary Information with Expanded View (EV) Figures and Tables that are collapsible/expandable online. A maximum of 8 EV Figures can be typeset. EV Figures should be cited as "Figure EV1, Figure EV2" etc... in the text and their respective legends should be included in the main text after the legends of regular figures.

5) a complete author checklist, which you can download from our author guidelines . Please insert information in the checklist that is also reflected in the manuscript. The completed author checklist will also be part of the RPF.

6) Please note that all corresponding authors are required to supply an ORCID ID for their name upon submission of a revised

manuscript (). Please find instructions on how to link your ORCID ID to your account in our manuscript tracking system in our Author guidelines

- the name of the statistical test used to generate error bars and P values,
- the number (n) of independent experiments (please specify technical or biological replicates) underlying each data point,
- the nature of the bars and error bars (s.d., s.e.m.),
- If the data are obtained from n {less than or equal to} 2, use scatter blots showing the individual data points.

12) All Materials and Methods need to be described in the main text using our 'Structured Methods' format, which is required for all research articles. According to this format, the Methods section includes a separate Reagents and Tools Table file (listing key reagents, experimental models, software and relevant equipment and including their sources and relevant identifiers) and a Methods and Protocols section describing the methods using a step-by-step protocol format. The aim is to facilitate adoption of the methodologies across labs. More information on how to adhere to this format as well as a downloadable template (.docx) for the Reagents and Tools Table can be found in our author guidelines: <https://www.embopress.org/page/journal/14693178/authorguide#structuredmethods>.

An example of a Method paper with Structured Methods can be found here: <https://www.embopress.org/doi/full/10.1038/s44320-024-00037-6#sec-4>

I look forward to seeing a revised form of your manuscript when it is ready.

Referee #1:

The manuscript by Alves et al reports that stabbing injury to the *Drosophila* adult brain causes an influx of calcium into glial cells, that activates DUOX, generating H₂O₂ that induces cell proliferation. The work demonstrates that an increase in ROS is essential to promote a regenerative response following injury to the *Drosophila* brain. The link between a surge in calcium, ROS, NADPH, H₂O₂ and JNK signalling in injury and regeneration had previously been reported in other animal model organisms and *Drosophila* contexts (eg labs of Amaya, Martin, Wood, eg Love et al 2013 *Nature Cell Biology*, Feng et al 2010 *PLoS Biology*, Weavers et al 2019 *Current Biology*, Razell et al 2013 *Current Biology*). This work compellingly demonstrates that *Drosophila* recapitulates events that are evolutionarily conserved across species and contexts (eg zebrafish, frogs, flies). The work will be of general interest and an important contribution to the fields of regenerative biology and brain plasticity, reinforcing the continued use of *Drosophila* as a model organism to investigate central nervous system injury, regeneration and repair. Overall, the work is of excellent quality. These experiments are difficult, given the natural variability in responses to injury and the time course of events (eg mitosis and apoptosis are fast processes both difficult to capture significantly). The work is generally rigorous and extensive, the evidence supports the claims and the manuscript is well written. Please allow me to make some suggestions that would significantly improve the work.

Major:

1. The work focuses on the involvement of glia in ROS production and cell proliferation after injury. However, upon *duox*-RNAi in glia upon injury, there is still a statistically significant increase in cell proliferation compared to non-injured controls. This could mean that neural progenitor cells also upregulate ROS and activate *duox*, which are not targeted with *repoGAL4*. In fact, an increase in ROS is required to activate neural stem cells and induce their proliferation (eg reviewed in Sies et al 2020 *Nat Rev Molec Cell Biol*). Since this lab showed that there are Dpn⁺ progenitor cells in the adult brain, they generated a tool to visualise them and demonstrated that they proliferate after injury (Simoes et al 2022 *Dev Cell*), they should test whether injury also induces ROS in Dpn⁺ cells and test if this leads to their activation and proliferation (eg with *dpnT2A>duoxRNAi*).
2. Page 4: "Among induced genes in regenerating cells (Sanchez et al., MS to be submitted), we also detected *jra/c-jun* and *atf-2*". And a selection of genes with expression up-regulated in glial cells is given in Figure 1M. However, as the RNAseq experiment was the foundation for choosing the genes for this project, then the RNAseq data must be included in this current paper, the Sanchez et al citation removed and in the Sanchez et al paper to be submitted soon they should cite this current paper instead. In brief, the cell sorting and RNAseq experiment should be mentioned more explicitly within the text and the total raw data of all differentially expressed genes, as well as other analyses (eg GO term, etc), should be provided as supplementary files with this paper.
3. The authors claim that the increase in calcium caused by injury originates from calcium entry through the plasma membrane, because knock-down of *serca* did not induce proliferation in the absence of injury (Figure S3). However, the median in *serca*RNAi is rather higher than in controls, and sample sizes are very different. If *n* increased, the difference could be statistically significant. To claim that calcium enters through a plasma membrane channel to lead to the activation of DUOX, the authors should knock-down a calcium membrane channel, like *cac*.
4. Data in Figure 2A-C and Figure 4 need to be quantified. Figure 4B-E: in the normal, non-injured brain both neurons and glia communicate via calcium signals (for glia, see Ma et al *Nature* 2016). GCaMP signals are reported as curves that spike and decay over time, over a background that normally already has GFP⁺ signal. Thus, average GCaMP6 signal intensity would be best measured across multiple flies without injury (controls), and before, during and after injury through time, to generate a GCaMP activation and decay curve. Figure 4F-G needs quantifying, like other data are.
5. Figure 3G-J. The analysis of the perma-twin clones requires lacks standard controls for clonal analyses: quantification of cell

number in non-injured control brains within K and L; and the percentage of brains with clones, both in injured and non-injured brains, is also missing. Also, the cell of origin in which the clones were generated (ie the genotype) must be provided. Having said that, perma-twin clones are based on the twin-spot MARCM technique, which relies on RNAi efficiently knocking down GFP and RFP in each twin clone. We have tried this method multiple times in our lab, with different genotypes, and including injury, and it produces many artifacts. It does not produce reliable evidence of cell proliferation nor cell fate and the paper would improve if these data were removed.

6. Based on the description in the methods, the statistical analysis would need revising. For example, Student t test is not suitable for data sets with more than 2 samples. Instead, for three or more samples, One Way ANOVA (normal distributions eg Figure 1G), Two-Way ANOVA (normal distributions, but comparing two variables, injury and RNAi, eg Figure 2P) or Kruskal-Wallis ANOVA (if distributions not normal) should be used instead. Figure 4O compares categories, and are thus categorical rather than numerical data (high, intermediate, low): X² or Kruskal Wallis ANOVA would be appropriate and the graph could be shown as percentages instead. The stats details should be provided in all graphs: ie test name, p value of whole data set test (eg p value for the One Way ANOVA test) plus multiple comparisons corrections tests (eg Dunnett to fixed control), to which the stars on the figures should refer. They also need to provide the results of comparisons not mentioned (eg is pH3 in pucRNAi statistically significantly different in injury vs non-injured brains?). The statistical details could be provided within the figure legends, within the text or as a separate document.

Minor points:

Literature citations could be improved to credit work that came before. For example, the previous findings from zebrafish, frogs, flies that calcium, ROS, H₂O₂, JNK are involved in regeneration - some cited - could be made more explicit; stabbing injury to the adult brain was first carried out by Kato et al 2009 Development; glial cell proliferation in response to injury was first reported by Kato et al 2011 PLoS Biology and Losada-Perez et al 2016 JCB.

Page 5: "according to transcriptional single cell resources" provide exact database name and URL.

Figure 2P: is the increase in proliferation in non-injured duoxRNAi statistically significantly different vs non-injured controls? If so, could authors provide an explanation?

Figure S2 F-G should show whole brain. Figure S2 would improve with quantifications for all data, as above. Figure S2M requires images.

Figure 2G-H: "Duox RNAi strongly apoptosis (TUNEL) (Figures 2G and 2H) compared to injured brains with intact Duox function, suggesting that Duox activation, but not dying cells per se, cause the strong increase in ROS in the injured brain area." It is not possible to rule out dying cells as the source of ROS, as they used tubGAL4, and dying cells continue to express genes, so they could have expressed duox-RNAi too.

Page 8 and Figure 3A,B experiment at 6{degree sign}C. This result did not reveal much, as temperature affects gene expression, so it would be best to remove it. It is quite a lateral experiment that was not pursued further, so removing it will not affect the rest of the study.

Figure 4F-G says CaMKII, but TRIC functions via calmodulin, not necessarily CaMKII. To verify the involvement of CaMKII, they could knock-down CaMKII after injury.

Figure 4P microscopy images should be included. The mycRNAi data are not relevant for this work and could be removed. The effect of pucRNAi should be compared to the non-injured control.

Genotypes are missing in many points (or only the parental stocks given in methods) and need to be included: e.g. genotype for cell sorting; genotype for TRIC experiment; clonal analysis genotype; details of RNAi lines. They could include a list of genotypes as a table or write full genotypes in figure legends.

Referee #2:

The manuscript by Alves et al. investigates the tissue regeneration mechanisms following a stab injury to the optic lobe (OL) of *Drosophila*. They have previously reported that the proliferation of glia and neural progenitor cells follows the injury as a regenerative response. In this manuscript, the authors aimed to dissect underlying mechanisms, utilizing numerous genetic tools. Overall, the work is interesting, based on their original model, and has broad implications for understanding tissue repair and regeneration. However, the manuscript as it is now does not entirely support the authors' conclusion. Listed below are the concerns that need to be addressed.

Major comments

1. Many figures of immunofluorescence results (which are most of the data in this paper) lack quantification. Most images are low magnification, which may be unavoidable given that they need to show a large part of the adult fly's OL. Nevertheless,

- increasing picture quality is recommended. More importantly, stating the expression levels go up or down without quantification diminishes credibility. This concerns Figs 2A-C, 2G-H, S2A-L, 4B-E, S3B-E'.
2. Figs. 2B-B", Tre-GFP and TUNEL staining. There is again no quantification, and no result of the pre-injury or control is shown.
 3. Fig 1M is a table listing some genes upregulated in glia following injury. However, more comprehensive data of the RNA-seq analysis is not included in the paper (unless I am mistaken).
 4. Based on the results of MitoTimer expression, the authors conclude that mitochondria are not involved in the oxidative stress response following the stab injury, despite the upregulation of ROS (page 5, paragraph 2). This is surprising and a key statement of the paper; however, as stated above, corresponding Figures (Figs 2A-C, S2A-S2E') do not include quantification results. Moreover, it is strange that, at least in the Figures, the GFP signal is barely visible. Non-oxidized mitochondria are marked by GFP, while no GFP signal suggests that MitoTimer is not properly expressed, rather than not being oxidized. This point has to be clarified.
 5. Figs 2G and H. The authors conclude that duox RNAi reduces ROS levels and gstG expression after injury as predicted; however, apoptosis (detected using the TUNEL staining) is not affected by reduced ROS. If this is the case, ROS does not contribute to cell death. To make this statement, quantification of the TUNEL staining signal is essential.
 6. Fig. S2M. Catalase or SOD2 overexpression is not sufficient to suppress injury-induced cell proliferation. This suggests that H₂O₂ is not a direct cause of the cell proliferation following injury. However, the authors reason that "probably due to insufficient suppression of the injury-induced ROS peak "(page 7, paragraph 1) without any supporting evidence. It is necessary to measure ROS levels in flies expressing catalase or SOD2 following injury to make this conclusion.
 7. Figs. 2M- P show that duox RNAi in all cells inhibits cell proliferation compared to non-injured flies and injured LacZ-overexpressed flies. However, in Fig. 2Q, while duox RNAi in glia reduces the number of proliferating cells following injury compared to the lacZ-expressing flies after injury, it does not fully suppress injury-induced cell proliferation. It means that there is a significant contribution from non-glia duox. The authors should acknowledge and discuss this fact.
 8. Figs. 3A and B. As stated in the manuscript (page 8, paragraph 1), GAL4 activity is temperature dependent, and it is not surprising that gstD-gal4 expression is greatly reduced at 60C. Therefore, to evaluate whether lower temperature has a protective effect after the stab injury, alternative methods to measure injury response, such as ROS levels and PH3 staining, should be used.
 9. Figs. 4F-H. It is puzzling that they used only the neuron-specific GAL4 driver to monitor calcium levels after injury, while the paper's focus is the response in glia. Why did they not use Repo-GAL4 >TRIC reporter to monitor calcium levels in glia after injury?
 10. Figs. 4K-O. How did the authors quantify TRE-GFP signal levels in elav+ and repo+ cells? Please describe more precisely the methods of these experiments.
 11. The paper's conclusion is "ROS are positive regulators of the regenerative response. They are important to keep elevated JNK activity, required for regeneration..." (page 10, last paragraph). While this is one way of looking at the results, I wonder why the injury itself is not considered. Injury causes ROS elevation (in glia and neurons), which triggers JNK signaling and cell proliferation as a regenerative response. Antioxidant treatment reduces JNK signaling and cell proliferation. But isn't it because the spreading of injury (cell death)-triggered ROS is reduced, and therefore regenerative response is reduced. If this is the case, the conclusion becomes somewhat trivial. In other words, is reducing ROS levels good or bad for the animal? Measuring the extent of cell death or other organismal consequences, like life span, may clarify this issue.

Minor comments

1. Page 4, paragraph 2. "As we previously found that glia are highly sensitive to injury". Does it mean glia die selectively than neurons?
2. Page 5, paragraph 2. "MitoTimer was activated with the optic lobe driver GMR-Gal4 prior to injury." What does "activated" mean? Wasn't it just expressed in the OL?
3. Was the efficiency of the UAS-duox RNAi tested? Multiple duox RNAi lines are available. How they elected to use this specific line should be described.

Referee #3:

In this manuscript, Rhiner and colleagues use the adult brain of *Drosophila* to identify roles of ROS and JNK in brain regeneration upon injury as visualized by pH3 (mitotic activity) staining. The paper is subdivided into 5 different chapters:

- (1) In the first chapter, authors show that JNK is activated in injured brain and this activation is accompanied by the production of ROS and partial lipid peroxidation, Control uninjured brain with TRE-GFP expression is lacking. Authors should provide evidence that JNK is being induced by other means (eg. pucker expression). Sentence of last para of pg 4 on the functional role of ROS in lipid peroxidation should be validated with functional data
- (2) In the second chapter, authors identify a role of Duox in ROS production and brain regeneration. Is Duox transcriptional induced by injury? pH3 staining: higher mag data should shown to demonstrate that these puncta are real mitotic figures. EdU incorporation is also a must in these experiments. Mito-Timer staining: high mag to see the organelles and the GFP/RFP ratio should be shown. Why is there no GFP staining in panels 2A-C? Memb localization of Duox in 2R-T is indeed of improvement. Add additional markers to verify membrane localization.
- (3) In the 3rd chapter, authors show that glial ROS (Duox activity) is required for injury-driven regeneration. The perma-twin labeling is also in need of improvement. Too many GFP, very low number of RFPs. What about twin clones? High mag should be shown and Ed U labeling is a must.

(4) In the 4th figure, authors show that Calcium influx as a result of injury induces Duxo-driven ROS. Functional data to conclude this are scarce. The role of JNK in regeneration should be validated by EdU experiments. Overall, the ms deals with an interesting topic, uses a great model system to do so, but at this stage is extremely preliminary and many of the conclusions need more convincing experimental data.

Point by point responses to Reviewer's comments

We thank the reviewers for their insightful and constructive comments and suggestions. We have tried to address all the concerns raised with additional experiments and improvements of Figures and MS text as pointed out by reviewers. These changes have enhanced the quality and understanding of the MS and we hope that the reviewers concur.

Please note that the MS has been shortened and reformatted in accordance with the *EMBOReport Reports* guidelines, in which Result and Discussion sections are merged. We have provided detailed information, where we have addressed points raised by the reviewers by providing page and paragraph numbers in blue that should facilitate the orientation in the adapted MS.

Referee#1:

The manuscript by Alves et al reports that stabbing injury to the *Drosophila* adult brain causes an influx of calcium into glial cells, that activates DUOX, generating H₂O₂ that induces cell proliferation. The work demonstrates that an increase in ROS is essential to promote a regenerative response following injury to the *Drosophila* brain. The link between a surge in calcium, ROS, NADPH, H₂O₂ and JNK signalling in injury and regeneration had previously been reported in other animal model organisms and *Drosophila* contexts (eg labs of Amaya, Martin, Wood, eg Love et al 2013 *Nature Cell Biology*, Feng et al 2010 *PLoS Biology*, Weavers et al 2019 *Current Biology*, Razell et al 2013 *Current Biology*). This work compellingly demonstrates that *Drosophila* recapitulates events that are evolutionarily conserved across species and contexts (eg zebrafish, frogs, flies). The work will be of general interest and an important contribution to the fields of regenerative biology and brain plasticity, reinforcing the continued use of *Drosophila* as a model organism to investigate central nervous system injury, regeneration and repair. Overall, the work is of excellent quality. These experiments are difficult, given the natural variability in responses to injury and the time course of events (eg mitosis and apoptosis are fast processes both difficult to capture significantly). The work is generally rigorous and extensive, the evidence supports the claims and the manuscript is well written. Please allow me to make some suggestions that would significantly improve the work.

We thank reviewer 1 for highlighting that the work will be an important contribution to the fields of regenerative biology and brain plasticity and that the work is overall of excellent quality.

Major:

1. The work focuses on the involvement of glia in ROS production and cell proliferation after injury. However, upon duox-RNAi in glia upon injury, there is still a statistically significant increase in cell proliferation compared to non-injured controls. This could mean that neural progenitor cells also upregulate ROS and activate duox, which are not targeted with repoGAL4. In fact, an increase in ROS is required to activate neural stem cells and induce their proliferation (eg reviewed in Sies et al 2020 *Nat Rev Molec Cell Biol*). Since this lab showed that there are Dpn+ progenitor cells in the adult brain, they generated a tool to visualise them and demonstrated

that they proliferate after injury (Simoes et al 2022 Dev Cell), they should test whether injury also induces ROS in Dpn+ cells and test if this leads to their activation and proliferation (eg with *dpnT2A>duoxRNAi*).

The contribution of *duox* function in progenitors is an interesting point to address as *repo-Gal4*-driven *duox*-RNAi does not completely suppress injury-induced proliferation. We tested this possibility, by conditional knock-down of *duox* with *dpnT2A-Gal4; tubGal80ts*. Unlike activation in glia, the results did not reveal a similar contribution of Duox function in neural progenitors. These results are shown in EV2M.

We therefore consider it likely that the remaining proliferation may stem from a minor role of progenitors and from glial subsets that show no or low expression of *repo* in the adult OL (Ferreira and Desplan, 2023).

We are discussing this possibility in the text on page.7, 1st paragraph.

2. Page 4: "Among induced genes in regenerating cells (Sanchez et al.,MS to be submitted), we also detected *jra/c-jun* and *atf-2*". And a selection of genes with expression up-regulated in glial cells is given in Figure 1M. However, as the RNAseq experiment was the foundation for choosing the genes for this project, then the RNAseq data must be included in this current paper, the Sanchez et al citation removed and in the Sanchez et al paper to be submitted soon they should cite this current paper instead. In brief, the cell sorting and RNAseq experiment should be mentioned more explicitly within the text and the total raw data of all differentially expressed genes, as well as other analyses (eg GO term, etc), should be provided as supplementary files with this paper.

We have improved this section on the bulk RNAseq data and apologize for the glitch that Table EV1 with the DE genes was not included in the submitted MS. We have removed the confusing comments on Sanchez et al., as still unpublished and mention exclusively changes in the glial RNAseq data from this study.

The results related to the bulk RNAseq are summarized on page 4, 1st paragraph. We also refer in the text to table EV1 containing all differentially regulated genes and have deposited the raw data under GEO accession number GSE309567 (access token: qfotwsqopnkbtx).

We also performed GO analyses and included it in EV1A.

Instead of the fold changes for DE genes discussed in the text, we generated dot plots (Figure 1N) and have highlighted the fold change and Padj value in table EV1 on the right side.

3. The authors claim that the increase in calcium caused by injury originates from calcium entry through the plasma membrane, because knock-down of *serca* did not induce proliferation in the absence of injury (Figure S3). However, the median in *serca*RNAi is rather higher than in controls, and sample sizes are very different. If *n* increased, the difference could be statistically significant. To claim that calcium enters through a plasma membrane channel to lead to the activation of DUOX, the authors should knock-down a calcium membrane channel, like *cac*.

We agree that the statement on calcium increase across the membrane was not sufficiently supported and the comparison with effects achieved with *serca*RNAi not convincing.

We thank the reviewer for the suggestion to look further into mechanisms and test involvement of the calcium membrane channel *Cac*. We managed to obtain a *cac* RNAi line, which on first order died at customs during the hot summer days. Upon reorder, we could repeat experiments with Gcamp signal recording before and after injury in combination with *cac* knock-down, which resulted in a significant suppression of glial calcium signal shown in Figure 4D- G. We therefore conclude that the calcium membrane channel *Cac* contributes to increased glial calcium levels.

We have removed the data on *serca* RNAi as the dataset was not extensive and the manipulation likely alters intracellular calcium with completely different kinetics (gradual increase with adult-onset induction versus acute injury) and therefore conclusions from comparing the two settings may not be valid.

4. Data in Figure 2A-C and Figure 4 need to be quantified. Figure 4B-E: in the normal, non-injured brain both neurons and glia communicate via calcium signals (for glia, see Ma et al Nature 2016). GCaMP signals are reported as curves that spike and decay over time, over a background that normally already has GFP+ signal. Thus, average GCaMP6 signal intensity would be best measured across multiple flies without injury (controls), and before, during and after injury through time, to generate a GCaMP activation and decay curve. Figure 4F-G needs quantifying, like other data are.

We have included the quantification of the experiments related to mitoTimer in Figure 2D and EV2D.

For GCaMP signal, we have followed the recommendations and have recorded average GCaMP6 signal intensity in multiple flies without injury (baseline GFP) and shortly after injury shown in Figure 4F. Recording the moment of injury could technically not been solved on multiple imaging set-ups without causing motion of sample as the objective interfered with the angle required for the lesions.

We therefore opted to shortly remove the preparation from the objective path, perform the lesion and immediately record the same brain after injury thereafter. The recordings were fused back-to-back for visualization (as described in the legends of Figure 4F and methods).

We have removed the images from previous Figure 4F-G (neuronal TRIC) as they were a more lateral finding and Reviewer 2 considered it more central to display TRIC signal in glia, which we have followed to improve clarity. The glial TRIC images and their quantification are displayed in Figure 4H-J.

5. Figure 3G-J. The analysis of the perma-twin clones requires lacks standard controls for clonal analyses: quantification of cell number in non-injured control brains within K and L; and the percentage of brains with clones, both in injured and non-injured brains, is also missing. Also, the cell of origin in which the clones were generated (ie the genotype) must be provided. Having said that, perma-twin clones are based on the twin-spot MARCM technique, which relies on RNAi efficiently knocking down GFP and

RFP in each twin clone. We have tried this method multiple times in our lab, with different genotypes, and including injury, and it produces many artifacts. It does not produce reliable evidence of cell proliferation nor cell fate and the paper would improve if these data were removed.

We are using here a modified version of the twin-spot MARCM with a continuous source of flippase as a complementary method to PH3. The system produces baseline signal under homeostatic conditions, which is expected over the extended period of activation (8 days with continuous Flippase expression). We have added the exact genotype of the analyzed flies to the Figure legend of 3G and the Method section.

The quantification of baseline signal as suggested is important and aids to appreciate the extent of injury-induced proliferation. We have recorded counts in non-injured controls on the respective diets, which are now shown in Figure 3G. All the counts above the homeostatic conditions are attributable to injury-induced proliferation, which is significant on control, but not on ROS quenching NAC diet. The results are discussed in the MS on page 8, 1st paragraph.

Regarding the clone retrieval frequency, almost 100% of brains showed a few labelled cells after 8 days of activation.

The tools may not be ideal in reporting absolute values, but it consistently reports relative differences in proliferation.

Twin-spot MARCM techniques are also used by other labs and detected e.g. increased glial proliferation in the VNC (+50% after injury) (Casas-Tinto et al., 2025).

6. Based on the description in the methods, the statistical analysis would need revising. For example, Student t test is not suitable for data sets with more than 2 samples. Instead, for three or more samples, One Way ANOVA (normal distributions eg Figure 1G), Two-Way ANOVA (normal distributions, but comparing two variables, injury and RNAi, eg Figure 2P) or Kruskal-Wallis ANOVA (if distributions not normal) should be used instead.

Figure 4O compares categories, and are thus categorical rather than numerical data (high, intermediate, low): X² or Kruskal Wallis ANOVA would be appropriate and the graph could be shown as percentages instead. The stats details should be provided in all graphs: ie test name, p value of whole data set test (eg p value for the One Way ANOVA test) plus multiple comparisons corrections tests (eg Dunnett to fixed control), to which the stars on the figures should refer. They also need to provide the results of comparisons not mentioned (eg is pH3 in pucRNAi statistically significantly different in injury vs non-injured brains?). The statistical details could be provided within the figure legends, within the text or as a separate document.

We thank the reviewer for pointing out improvements and have revised the statistical methods and provide the full statistical details (SD, Test used, p values, N numbers) within the Figure legends for each graph in accordance with normal or non-normal distributions.

Figure 4O (new Figure 4R): We have changed the graph 4O to display information on the relative abundance of categories within the JNK positive cells instead of the numerical data, which was indeed a better way to display the results. We do not compare the distribution to other timepoints, but rather capture the distribution of JNK^{low} vs JNK^{high} in glia versus neurons. Therefore, no further statistics was applied. The respective stainings used to identify glia (Repo) and neurons (Elav) are now shown in Figure EV4H.

Minor points:

Literature citations could be improved to credit work that came before. For example, the previous findings from zebrafish, frogs, flies that calcium, ROS, H₂O₂, JNK are involved in regeneration - some cited - could be made more explicit; stabbing injury to the adult brain was first carried out by Kato et al 2009 Development; glial cell proliferation in response to injury was first reported by Kato et al 2011 PLoS Biology and Losada-Perez et al 2016 JCB.

ROS and Duox play highly conserved functions in diverse models. We have cited relevant findings from mouse, Zebrafish and fly studies, but could not cover all interesting systems and findings in this report.

We are aware of the pioneering Kato et al. 2009 paper that first reported glial proliferation in the adult brain and have included it, along other important work on plasticity in the adult nervous system. We had cited it in previous publications when we introduced the OL lesion paradigm, but agree that providing further context of relevant work was also indicated here.

We have included references on relevant work on adult brain plasticity or conserved signalling on Page 3

In addition, we added that glial cell proliferation also occurs in the developing VNC upon injury (Kato et al 2011) as a further interesting setting.

Page 5: "according to transcriptional single cell resources" provide exact database name and URL.

We have included the link to the database (NCBI-GEO: GSE142787) on page 5, last paragraph.

Figure 2P: is the increase in proliferation in non-injured duoxRNAi statistically significantly different vs non-injured controls? If so, could authors provide an explanation?

The increase is not statistically significant. We have added this information to the graph in Figure 2S (new).

Figure S2 F-G should show whole brain. Figure S2 would improve with quantifications for all data, as above. Figure S2M requires images.

We have extensively quantified samples for previous Figure S2F,G with the ratiometric hydrogen peroxide-specific ROS reporter and improved analyses and display.

The dot plot has been integrated in Figure 2K and the images in EV2H,I.

For Figure S2/EV2, we have included quantification for EV2A-C in EV2D for mitoTimer and EV2L for *hmlGal4*, *UASgfp* images. For PH3 staining (read-out of S2M), we have included OL images in Figure 2P-R.

Figure 2G-H: "Duox RNAi strongly ... apoptosis (TUNEL) (Figures 2G and 2H) compared to injured brains with intact Duox function, suggesting that Duox activation, but not dying cells per se, cause the strong increase in ROS in the injured brain area." It is not possible to rule out dying cells as the source of ROS, as they used tubGAL4, and dying cells continue to express genes, so they could have expressed duox-RNAi too.

This statement has been modified as it was unclear. We wanted to show lesions with a comparable amount of apoptosis with and without *duox* function in the tissue to illustrate the impact on the ROS-responsive *gstD-gfp* reporter.

We have modified the text to:

we further knocked-down duox prior to injury with the above tools, but in combination with the gstD-GFP reporter. Duox RNAi strongly reduced oxidative stress compared to control or nox RNAi for lesions that caused a comparable amount of apoptosis (TUNEL) (Figures. 2I, J and EV2G). We included a representative image of crosses with *nox* RNAi as a further comparison.

Page 8 and Figure 3A,B experiment at 6{degree sign}C. This result did not reveal much, as temperature affects gene expression, so it would be best to remove it. It is quite a lateral experiment that was not pursued further, so removing it will not affect the rest of the study.

True, we have removed the Figure and text related to this lateral experiment.

Figure 4F-G says CaMKII, but TRIC functions via calmodulin, not necessarily CaMKII. To verify the involvement of CaMKII, they could knock-down CaMKII after injury.

We have modified and mention the tool by its known short form TRIC.

Figure 4P microscopy images should be included. The *myc*RNAi data are not relevant for this work and could be removed. The effect of *puc*RNAi should be compared to the non-injured control.

We have removed the *myc*RNAi data in now Figure 4S as they are not directly relevant here and show all comparisons to controls.

Due to space constraints, we have included microscopy images where the read-out (PH3) is first used (Figure 2P-R) and show full OL images instead of insets as requested in other comments.

Genotypes are missing in many points (or only the parental stocks given in methods) and need to be included: e.g. genotype for cell sorting; genotype for TRIC experiment; clonal analysis genotype; details of RNAi lines. They could include a list

of genotypes as a table or write full genotypes in figure legends.

We have included the genotypes wherever missing in the images in the Figure legends and Methods section, including details on the RNAi lines in the comprehensive Methods and Tools table.

Referee#2:

The manuscript by Alves et al. investigates the tissue regeneration mechanisms following a stab injury to the optic lobe (OL) of *Drosophila*. They have previously reported that the proliferation of glia and neural progenitor cells follows the injury as a regenerative response. In this manuscript, the authors aimed to dissect underlying mechanisms, utilizing numerous genetic tools. Overall, the work is interesting, based on their original model, and has broad implications for understanding tissue repair and regeneration. However, the manuscript as it is now does not entirely support the authors' conclusion.

Listed below are the concerns that need to be addressed.

We thank reviewer 2 for stating that the work is interesting and has broad implications for understanding tissue repair and regeneration.

Major comments

1. Many figures of immunofluorescence results (which are most of the data in this paper) lack quantification. Most images are low magnification, which may be unavoidable given that they need to show a large part of the adult fly's OL. Nevertheless, increasing picture quality is recommended. More importantly, stating the expression levels go up or down without quantification diminishes credibility. This concerns Figs 2A-C, 2G-H (statement), S2A-L, 4B-E (Videos and Curves), S3B-E'.

We have performed and included quantifications for almost all the images shown. Statements with respect to increase or decrease of expression are now extensively supported by quantification and statistical analyses included in the Figure legends.

In particular, we have included a graph for Figure 2A-C in Figure 2D, a graph for S2A-C in Figure EV2D, graphs for Figures S2D-G in Figure 2E and 2K, graphs for S2H,I in Figure EV2L and curves for Figure 4B-E in Figure 4F.

For Figure 2G,H, we have included a representative image of an additional experiment (RNAi of *nox*) in EV2G and show lesions with comparable amount of apoptosis with a clear differential effect on the ROS responsive reporter *gstD-GFP*.

2. Figs. 2B-B", Tre-GFP and TUNEL staining. There is again no quantification, and no result of the pre-injury or control is shown.

We understand the comment refers to Figure 1B and we have included a pre-injury image in Figure 1B. TRE-GFP is exclusively induced in the lesioned OL and there is no signal in the contralateral lobe. We have described the effect of

the lesion on apoptosis (injured and non-injured OLs) in Simoes et al., 2022 Dev Cell and Moreno et al., 2015 Curr Biol.

3. Fig 1M is a table listing some genes upregulated in glia following injury. However, more comprehensive data of the RNA-seq analysis is not included in the paper (unless I am mistaken).

We have included more comprehensive data of the RNAseq analysis and apologize for the oversight that a table containing the DE genes was not included in the MS.

The results related to the bulk RNAseq are summarized on page 4 , 1st paragraph. We now refer in the text to table EV1 containing all differentially regulated genes and have deposited the raw data under GEO accession number GSE309567 (access token: qfotwsqopnkbtx).

We also included a Figure on GO terms in EV1A.

Instead of the fold changes for DE genes discussed in the text, we generated dot plots (Figure 1N) and have highlighted the fold change and Padj value in table EV1 on the right side.

4. Based on the results of MitoTimer expression, the authors conclude that mitochondria are not involved in the oxidative stress response following the stab injury, despite the upregulation of ROS (page 5, paragraph 2). This is surprising and a key statement of the paper; however, as stated above, corresponding Figures (Figs 2A-C, S2A-S2E') do not include quantification results. Moreover, it is strange that, at least in the Figures, the GFP signal is barely visible. Non-oxidized mitochondria are marked by GFP, while no GFP signal suggests that MitoTimer is not properly expressed, rather than not being oxidized. This point has to be clarified.

The images shown in Figs 2A-C, S2A-S2E represented a merge of all channels and GFP signals could not be seen, although it was present.

We are now showing the GFP and RFP signal of OLs separately, in addition to the merged images.

Moreover, we have included quantifications for the mitoTimer experiments (Figure 2D and Figure EV2D) that did not reveal any statistically significant change.

In addition, we corroborated results with a second, highly sensitive mitochondrial ROS sensor. Quantifications did not detect a notable contribution of mitochondria after stab lesions and results are shown in Figure 2E and Figure EV2E,F.

5. Figs 2G and H. The authors conclude that duox RNAi reduces ROS levels and gstG expression after injury as predicted; however, apoptosis (detected using the TUNEL staining) is not affected by reduced ROS. If this is the case, ROS does not contribute to cell death. To make this statement, quantification of the TUNEL staining signal is essential.

This statement has been modified as it was unclear. We wanted to show lesions with a comparable amount of apoptosis with and without *duox* function in the tissue to illustrate the impact on the ROS-responsive *gstD-gfp* reporter.

We have modified the text to:

we further knocked-down duox prior to injury with the above tools, but in combination with the gstD-GFP reporter. Duox RNAi strongly reduced oxidative stress compared to control or nox RNAi for lesions that caused a comparable amount of apoptosis (TUNEL) (Figures. 2I, J and EV2G). We included a representative image of crosses with nox RNAi as a further comparison.

6. Fig. S2M. Catalase or SOD2 overexpression is not sufficient to suppress injury-induced cell proliferation. This suggests that H₂O₂ is not a direct cause of the cell proliferation following injury. However, the authors reason that "probably due to insufficient suppression of the injury-induced ROS peak "(page 7, paragraph 1) without any supporting evidence. It is necessary to measure ROS levels in flies expressing catalase or SOD2 following injury to make this conclusion.

As we did not detect an effect, we speculate that NAC and RNAi of *duox*, which quenches ROS at the source are more efficient in suppressing the ROS-dependent response, but we have no quantitative data for ROS levels. From other genetic experiments in the injured wing disc, it appears that a recombinant fly *UAS-Sod1:UAS-Cat (Sod1:Cat)* construct is more efficient at suppressing ROS, resulting in striking effects only when both components are overexpressed (Santabárbara-Ruiz et al., 2015, PLOS Genetics). The authors state: *In the presence of hydrogen peroxide, Catalase (Cat) catalyzes its breakdown into water and oxygen. Thus, overexpression of Sod or Cat will remove their respective ROS substrates, whereas simultaneous activation of Sod and Cat will enhance the depletion of both O₂⁻ and H₂O₂.*

We are mentioning this as a possible explanation in the text on page 8, end of 2nd paragraph.

7. Figs. 2M- P show that *duox* RNAi in all cells inhibits cell proliferation compared to non-injured flies and injured LacZ-overexpressed flies. However, in Fig. 2Q, while *duox* RNAi in glia reduces the number of proliferating cells following injury compared to the lacZ-expressing flies after injury, it does not fully suppress injury-induced cell proliferation. It means that there is a significant contribution from non-glial *duox*. The authors should acknowledge and discuss this fact.

We mention in the text that glial *duox*-RNAi does not completely suppress injury-induced proliferation. We tested the possibility that progenitors may be involved, by scoring proliferation with conditional knock-down of *duox* with *dprT2A-Gal4; tubGal80ts*. Unlike activation in glia, the results did not reveal a similar contribution of *Duox* function in neural progenitors. These results are shown in EV2M.

We therefore consider it likely that the remaining proliferation may stem from a minor role of progenitors and from glial subsets that show no or low expression of Repo in the adult OL (Ferreira and Desplan, 2023).

We are discussing this possibility in the text on page.7, 1st paragraph.

8. Figs. 3A and B. As stated in the manuscript (page 8, paragraph 1), GAL4 activity is temperature dependent, and it is not surprising that *gstD-gal4* expression is greatly reduced at 6°C. Therefore, to evaluate whether lower temperature has a protective effect after the stab injury, alternative methods to measure injury response, such as ROS levels and PH3 staining, should be used.

This experiment was confusing as we intended to make the point that there is a confounding effect of temperature on reporter expression to motivate the choice of chemical ROS suppression. We have removed the Figure and text regarding this lateral experiment.

9. Figs. 4F-H. It is puzzling that they used only the neuron-specific GAL4 driver to monitor calcium levels after injury, while the paper's focus is the response in glia. Why did they not use *Repo-GAL4 >TRIC* reporter to monitor calcium levels in glia after injury?

We agree that showing results with *Repo-Gal4>TRIC* was more fitting and indicated in this Figure and have included these findings in Figure 4H-J, replacing the panels with *nsyb-Gal4*,

10. Figs. 4K-O. How did the authors quantify TRE-GFP signal levels in *elav+* and *repo+* cells? Please describe more precisely the methods of these experiments.

The quantification was performed on TRE-GFP expressing brains that were stained for the glial marker *Repo* and neuronal marker *Elav*. We have included images that show all channels in EV4G,H and included text in the Methods section.

11. The paper's conclusion is "ROS are positive regulators of the regenerative response. They are important to keep elevated JNK activity, required for regeneration..." (page 10, last paragraph). While this is one way of looking at the results, I wonder why the injury itself is not considered. Injury causes ROS elevation (in glia and neurons), which triggers JNK signaling and cell proliferation as a regenerative response. Antioxidant treatment reduces JNK signaling and cell proliferation. But isn't it because the spreading of injury (cell death)-triggered ROS is reduced, and therefore regenerative response is reduced. If this is the case, the conclusion becomes somewhat trivial. In other words, is reducing ROS levels good or bad for the animal?

Measuring the extent of cell death or other organismal consequences, like life span, may clarify this issue.

The main contribution of ROS in this injury paradigm is mediated by Duox activation in glia as we found here, which release extracellular ROS that diffuse in the injured brain area. Indeed, ROS elevation per se cause some JNK activation. However acutely after injury, JNK is also activated due to mechanical stress (cell tearing, stretching and compression by stab lesion), which can activate JNK via alternative routes e.g. FAK/Src and small GTPases.

Antioxidant treatment suppresses the local injury-induced ROS but has no effect on the mechanical stress. We therefore examined the effect on JNK pathway activity several days after the acute impact in order to evaluate how JNK signalling is sustained by ROS in numerous cells at the injury site.

The question if reducing ROS is good or bad at the organismal level is complex as ROS generation may be useful for brain repair upon injury and infection, but comes at the price of causing increased tissue inflammation with aging.

We have measured lifespan and general resilience to brain injury in flies with tub-Gal4-driven *duox* RNAi. The results showed that flies with reduced Duox function show a significant increase in mean lifespan, which matches results reported for *duox* heterozygosity (Baek et al., 2022). Results were similar for both tested RNAi lines. In combination with our injury, performed with a thin sterile filament, we did not find significant differences in mean survival although there was some trend for increased mortality shortly after injury.

We conclude that “further work involving *duox* manipulation in glia, combined with brain function assays will be required to better understand how Duox-dependent plasticity influences brain repair and behavior.

.”

We have included a discussion of these questions on page 10, second last paragraph.

Minor comments

1. Page 4, paragraph 2. "As we previously found that glia are highly sensitive to injury". Does it mean glia die selectively than neurons?

No, the sentence should have expressed “responsive to injury”. We have modified this sentence and now write: As we previously found that glia act as important coordinators of neuro-glial interactions, Page 4, 1st sentence.

2. Page 5, paragraph 2. "MitoTimer was activated with the optic lobe driver GMR-Gal4 prior to injury." What does "activated" mean? Wasn't it just expressed in the OL?

Yes, we have corrected it.

3. Was the efficiency of the UAS-*duox* RNAi tested? Multiple *duox* RNAi lines are available. How they elected to use this specific line should be described.

We selected a *duox* RNAi for which the efficiency to suppress ROS has been shown (BL32903) (Kizhedathu et al., Elife 2021), which we also confirmed in our experiments. We found that both BL32903 and BL32907 behave similar in life span experiments and both lines also showed efficient ROS suppression and similar phenotypes in the gut (Liu Z, Zhang H, Lemaitre B, Li X, Cell Reports 2024).

We have specified in the Reagent and Tools table which *duox* RNAi line was used for most experiments and have added an explanation to the text, page 5, last paragraph.

Referee

#3:

In this manuscript, Rhiner and colleagues use the adult brain of *Drosophila* to identify roles of ROS and JNK in brain regeneration upon injury as visualized by pH3 (mitotic activity) staining. The paper is subdivided into 5 different chapters: (1) In the first chapter, authors show that JNK is activated in injured brain and this activation is accompanied by the production of ROS and partial lipid peroxidation, Control uninjured brain with TRE-GFP expression is lacking. Authors should provide evidence that JNK is being induced by other means (eg. pucker expression).

We understand that first chapter refers to Figure 1B and have included a control non-injured brain in Figure 1B. TRE-GFP is exclusively induced in the lesioned OL and there is no signal in the contralateral lobe.

We previously showed by immunohistochemistry that stab lesions trigger upregulation of MMP1 (Simões et al., 2022 Dev Cell) and show here transcriptional activation of *mmp1* in Figure 1N. *mmp1* is a highly conserved direct downstream target activated by JNK signalling and brain injury. MMP1 and AP-1-motif containing genes are also highly induced in a closed head model of brain injury (Byrns et al., Nat Aging. 2021).

Sentence of last para of pg 4 on the functional role of ROS in lipid peroxidation should be validated with functional data.

We have used the ROS-generating agent paraquat to functionally show that increased ROS cause lipid peroxidation with the tools used.

We have modified the conclusions of this part, which may have been too wide to “lipid peroxidation in injured OLs suggested that the activated anti-oxidant responses are efficient in restricting ROS damage” on page 4, second last sentence.

(2) In the second chapter, authors identify a role of Duox in ROS production and brain regeneration.

pH3 staining: higher mag data should shown to demonstrate that these puncta are real mitotic figures.

Cells in the brain are very tightly packed. We have included images that capture some mitotic figures in Figure for Reviewer 3.

EdU incorporation is also a must in these experiments.

We have assessed proliferation with two different approaches: (i) PH3 staining, which is specific for mitotic cells in contrast to EdU and (ii) mitotic-dependent

labelling and the request to repeat all time-consuming proliferation assays with less suitable EdU are not justified as they would require another year of work and are beyond the scope of a compact EMBO Report.

We have previously shown that injury to the OL also leads to EdU incorporation in cells at the lesion site (48h after feeding mostly glia) and with longer feeding, EdU signal can also be found in newly formed neurons (Elav+,EdU+) (Simões et al., 2022; Fernandez-Hernandez et al., 2013 Supplementary Data).

However, prolonged feeding of EdU to adult flies is quite toxic, likely due to EdU incorporation into DNA, which has been described.

It leads to high mortality of injured flies and great variability in obtained results, which make it unsuitable for screening and comparing different genetic manipulations.

We have included some images of retrieved labelling in Figure for Reviewer 3. Extensive optimizations of the protocol may eventually allow for reliable quantifications, but could not be achieved within the revision period.

Mito-Timer staining: high mag to see the organelles and the GFP/RFP ratio should be shown.

We show higher magnification images of mitoTimer and quantifications of the GFP/RFP ratio in Figure 2A-C, EV2A-C with quantifications of the GFP/RFP ratios in Figure D and EV2D.

We also show high magnification images for the mitochondrial targeted ROS reporter in Figure EV2E,F and quantifications in Figure 2E.

Why is there no GFP staining in panels 2A-C?

The GFP channel could not be sufficiently appreciated in the merged images We have changed the display and now show OL images showing the green, red and merged channel with DAPI in Figure 2A-D and EV2A-D.

Memb localization of Duox in 2R-T is indeed of improvement.

The construct has been previously shown to localize to the membrane. We have activated it at adult stage and verified membrane localization by showing colocalization with the membrane marker E-cadherin, shown now in Figure EV2N.

(3) In the 3rd chapter, authors show that glial ROS (Duox activity) is required for injury-driven regeneration. The perma-twin labeling is also in need of improvement. Too many GFP, very low number of RFPs. What about twin clones? High mag should be shown and EdU labeling is a must.

The clone appearance with the perma-twin system has been described and quantified in Fernández-Hernández et al., 2013 including insets, revealing symmetric divisions.

The red membrane signal is typically more difficult to see than the green signal.

(4) In the 4th figure, authors show that Calcium influx as a result of injury induces Duxo-driven ROS. Functional data to conclude this are scarce.

We have improved data showing that acute injury leads to increased calcium levels in glia by quantifying GCaMP signal before and after injury and by analyzing TRIC in glia (Figure 4B-J). We also provide additional functional evidence that the membrane calcium channel (Cac subunit) is involved in this response.

Findings from the Wood and Martin lab have previously shown that Duox-dependent ROS generation is dependent on the Duox intracellular calcium-sensing domain.

The role of JNK in regeneration should be validated by EdU experiments.

We have assessed proliferation with two different approaches: (i) PH3 staining, which is specific for mitotic cells in contrast to EdU and (ii) mitotic-dependent labelling and the request to repeat all time-consuming proliferation assays with less suitable EdU are not justified as they would require another year of work and are beyond the scope of a compact EMBO Report.

Overall, the ms deals with an interesting topic, uses a great model system to do so, but at this stage is extremely preliminary and many of the conclusions need more convincing experimental data.

We thank Reviewer 3 for stating that the MS deals with an interesting topic and uses a great model system. We do not consider the results preliminary as they have been corroborated with different methods and tools and we have performed extensive quantifications to support the conclusions.

Dear Dr. Rhiner,

Thank you for your patience while your revised manuscript was re-reviewed. It was sent to all 3 referees but unfortunately, 2 of them were not available to re-review your study. I therefore asked referee 1 to please assess your response to all referee comments.

We have now received the enclosed reports from referee 1, who still has a few more minor suggestions that I would like you to address and incorporate before we can proceed with the official acceptance of your manuscript. Please co-submit a point-by-point response to all remaining comments with your final ms.

A few editorial requests will also need to be addressed:

- Please reduce the number of keywords to 5.
- Please move the Disclosure statement to after the Acknowledgments. The Figure legends should be placed after the References, at the very end of the ms.
- There is one author name discrepancy - Andreia Augusto in the ms file vs. Andreia Augusta in the online ms submission system, please correct.
- If Champalimaud Research is a biotech company, employment in a biotech company should be stated in the Disclosure statement.
- The authors credits need to be removed from the ms file. All credits need to be entered during online ms submission.
- The REFERENCE format is not correct, et al needs to be used after 10 author names. Please use the EMBO reports reference style.
- The FUNDING INFO needs to be congruent in the ms file and in our online submission system. This seems to be missing in the system: the FCT ERC-Portugal program and the Champalimaud Foundation and facility funding by CONGENTO LISBOA-01-0145-FEDER-022170. The Comments box should not be used for listing funders, but the separate entries (More Funders option) should be used.
- Table EV1 needs to be updated to Dataset EV1 as it is more complex than a table. The title needs to be corrected in all places to Dataset EV1 (source file name, title in the system, callout in the ms).
- The Reagents & Tools TABLE needs to be removed from the ms file and uploaded as a separate file.
- Our systematic image analysis of all to-be-accepted ms detected a cell reuse between Figure 1J and Figure 3C that is not listed in the figure legends. Please explain/clarify.

* Figure Legends - Comments *

- Please note that the legend for figure 1 is not provided in the sequential manner. This needs to be rectified.
- Please define the annotated p values ****/***/**/* as well as provide the exact p-values for the same in the legend of figure 1N as appropriate.
- Please note that the exact p values are not provided in the legends of figures 1H, I, U; 2D, H, K, S, T; 3D, G, J; 4F, J, M, S, T; EV1 H, EV2 D, M; EV3 A, EV4 C, F, I; please provide exact p-values as reasonable.
- Please note that the error bars are not defined in the legends of figures 2E, K
- Please indicate the statistical test used for data analysis in the legends of figures 1N, 2E, 4G
- Please note that the dotted borders are not defined in the legend of figure 2U', 4K, L, N, O, P, Q; EV2 A', A'B', B', C', C', EV4 A', B', D', E' . This needs to be rectified.
- Please note that the dotted lines are not defined in the legend of figures 1C, E, F, G, K, L, M; 2M, N, Q, R, 3B, C, E, F, H, I; 4C, C', E, E', I, I', O-Q; EV2 K, EV4 H. This needs to be rectified.
- Please note that the dotted boxes are not defined in the legend of figures 1F, G; 2U'; 3E, F, H, I; EV2 G. This needs to be rectified"

EMBO press papers are accompanied online by A) a short (1-2 sentences) summary of the findings and their significance, B) 2-3 bullet points highlighting key results and C) a synopsis image that is exactly 550 pixels wide and 200-600 pixels high (the height is variable). The synopsis image should provide a sketch of the major findings, like a graphical abstract. Please note that

text needs to be readable at the final size. Please send us this information along with the final manuscript.

Referee #1:

Reviewer 1 Revision of revised version:

The authors have made a good effort to improve the manuscript. I am satisfied by their response to the criticisms I had raised, as - on the whole - the new experiments, quantifications and analyses resolved those criticisms. I am satisfied that no further experimental revisions are required to enable publication of the work.

The authors could pay some attention to some minor points below, when polishing the final version of the text.

On the contribution of duox in neural stem cell progenitors to the injury response, the authors wrote:

"The contribution of duox function in progenitors is an interesting point to address as repo-Gal4-driven duox-RNAi does not completely suppress injury induced proliferation. We tested this possibility, by conditional knock-down of duox with *dpnT2A-Gal4*; *tubGal80ts*. Unlike activation in glia, the results did not reveal a similar contribution of Duox function in neural progenitors. These results are shown in EV2M. We therefore consider it likely that the remaining proliferation may stem from a minor role of progenitors and from glial subsets that show no or low expression of repo in the adult OL (Ferreira and Desplan, 2023). We are discussing this possibility in the text on page.7, 1st paragraph".

Reviewer's response:

The additional experiment in EV2M for the revised version is nice, although the images on which the graph data are based ought to be provided. The data in EV2M show that upon injury, the number of pH3+ mitotic cells in the brain increased in controls (*lacZ*), and there was no significant difference between injured controls and injured brains with *duoxRNAi* in *dpn+* NSCs. The authors interpret that NSC do not contribute to duox-induced proliferation after injury. However, an alternative interpretation is that since presumably there are more glial cells than *Dpn+* progenitors, any effect by *Dpn+* cells would be diluted. Furthermore, when comparing *tubGAL80ts*, *dpn>duoxRNAi* injured vs not-injured, there was no significant difference either, meaning that duox is required in NSC for injury-dependent mitosis. If duox were not required for injury-dependent proliferation of NSCs, then overall mitosis in injured *duoxRNAi* samples should have been significantly different from that of non-injured *duoxRNAi* controls. In fact, when they knocked-down duox in glia, there was a significant difference between injured vs non-injured *duoxRNAi* samples (Figure 2T), which had prompted the comment (raised by two reviewers) that glia could not be the only source of duox after injury. Thus, the authors' interpretation of data - ie that glia are required for duox-dependent injury-induced proliferation whereas *dpn+* cells are not - is not consistent with the data shown. Instead, the data show that both glia and *dpn+* cells are involved in duox-dependent cell proliferation after injury. This makes more sense also with what is known about how ROS induces proliferation of NSC and with previous published work by these same authors that NSCs proliferate after injury in *Drosophila*.

The authors response including additional data and analyses to my points 2, 3 and 4 are excellent, thank you.

However, I am afraid the authors did not resolve point 5 on perma-twin-clones. Unfortunately, twin-spot MARCM (Yu et al 2009) and Perma-Twin (Fernandez-Hernandez et al, 2013) clones based on twin-spot MARCM with actin driven UAS-FLP - both methods - produce abundant artefacts that prevent lineage tracing. The fact that some authors continue to use these methods does not make artefacts disappear nor the science any better. Rather, it is worrying if the community perpetuates artefacts. For example, in Figure 3E,F,H,I the number of RFP+ and GFP+ cells differs greatly, with many more red cells than green cells (their quantifications ought to have been given separately in the graphs). MARCM clones require cell proliferation to occur. Generally, glial cells divide symmetrically, and if glial cells were the major contributors to injury-induced proliferation as this paper proposes, then the resulting twin red and green clones should contain equal number of glia cells, clustering together as glia normally do, and would be expected to have characteristic glial shape. But the images do not show such clones. Furthermore, the controls Figure 3E,H differ greatly. Potentially, the reason these cell division do not result in the expected progeny cells and clones is because the micro-RNAs in twin-spot MARCM do not cause reliable, stable, persistent knock-down of RFP and GFP over time, and GFP and RFP are expressed regardless of lineage. The effects of NAC and VitC are significant, so if they are to be kept, then Figure 3E-J should be moved to supplementary figures, GFP and RFP cell counts should be provided separately, and an explanation of the caveats of the method should be included within the main text. Having said that, the manuscript contains

abundant, high-quality data, thus I strongly advice removing the perma twin MARCM data (Figure 3E-J). Removing these data (Figure 3E-J) would improve the quality of the paper, the science and the consistency of the conclusions.

Minor:

Figure 1N: figure legend needs detail on what the Y-axis values mean in this graph.

Figure 3G,J GFP and RFP cell numbers should be given separately. Also, images from non-injured controls must be provided.

Figure 2T, Figure 4S and Figure EV3, Figure EV2M image data should be provided.

In the Methods section, the description of conditional expression in experiments using tubGal80ts, including the perma-twin MARCM, is missing. These methods were used for data in Figures 2,3,4. For example, at what temperature were flies raised, when were they shifted to what temperature, and were they kept at that temperature until dissection (and any variation across experiments)?

Statistical analysis has improved, but there is still room for improvement. For example, Figure 2D legend says Kruskal Wallis and Dunn test were used. However, Dunn test is to a fixed control (ie the same control for all comparisons), and instead, the graph shows comparisons of everything against everything. Figures 2K, S, T seem to require Two Way ANOVA instead. I recommend the authors seek expert advice. Most likely it will not make a difference to the conclusions, as overall data are clear and sample sizes large enough.

Requested feedback on the authors' response to Reviewer 3's review:

Reviewer 3 comments are included within "..".

(1) "In the first chapter, authors show that JNK is activated in injured brain and this activation is accompanied by the production of ROS and partial lipid peroxidation. Control uninjured brain with TRE-GFP expression is lacking. Authors should provide evidence that JNK is being induced by other means (eg. puckered expression)."

This was a valid point, and the authors responded appropriately, providing the required image and data showing that the JNK target MMNP1 is up-regulated upon injury.

"Sentence of last para of pg 4 on the functional role of ROS in lipid peroxidation should be validated with functional data". It is unclear what further evidence Reviewer 3 expected here. Based on this ambiguity, it would appear the authors responded appropriately.

(1) "pH3 staining: higher mag data should shown to demonstrate that these puncta are real mitotic figures"

This was a valid point. The Figure for Reviewer 3 is good and could be included in the paper as a supplementary figure, together with similar images for the other genotypes too (eg in Figure 2P,Q,R). The larger images in the revised version are also nicer than the original, as it is clearer to see increase in pH3+ spots at the lesion site.

"EdU incorporation is also a must in these experiments".

I disagree with this reviewer. EdU (like BrdU) label cells in S-phase. However, cells can arrest in G2, so going through S phase does not guarantee that cells will divide. In fact, EdU and BrdU are also used as evidence for polyploidy. EdU and BrdU labelling have been used in the past a lot as evidence of cell cycling because they are considerably easier to do experimentally than using mitotic markers, because S-phase lasts much longer than mitosis. Furthermore, EdU and BrdU intercalate into DNA and the label remains even after S-phase has finished and throughout subsequent divisions. Anti-pH3 is a mitotic marker instead, and more accurate evidence of cells undergoing cell division. Thus, in my view the authors' response was appropriate.

"Mito-Timer staining: high mag to see the organelles and the GFP/RFP ratio should be shown"

This was a fair point. The authors have improved the presentation of these data, both images and quantification.

"Why is there no GFP staining in panels 2A-C?"

This was a valid point. The authors have responded appropriately, by showing the split channels.

"Memb localization of Duox in 2R-T is indeed of improvement"

This was a critical point for the authors' interpretation. The images provided in Figure EV2N are not convincing, and the genotype does not seem right. If it is tubGAL4, why is expression only switched on in clusters of cells, shouldn't it come on in all cells? In any case the images are not convincing because the red signal (duox-mruby2) does not coincide with the green signal (Ecahderin) at membranes. Figure 2U-W, present in the original submission, was better than the response. Thus, this point was not improved.

(2) "In the 3rd chapter, authors show that glial ROS (Duox activity) is required for injury driven regeneration. The perma-twin

labeling is also in need of improvement. Too many GFP, very low number of RFPs. What about twin clones? High mag should be shown and EdU labeling is a must".

This was a valid point. Although I do not agree that EdU is a must for the reasons given above, I do agree with the criticisms of the perma twin clones, which I had also raised. The authors have not resolved these criticisms.

(3) "In the 4th figure, authors show that Calcium influx as a result of injury induces Duxo-driven ROS. Functional data to conclude this are scarce."

This was a valid point and was addressed appropriately by the authors.

"The role of JNK in regeneration should be validated by EdU experiments".

This criticism is not valid. The authors had already shown in the original submission that loss of function of puc prevented injury-induced cell proliferation, as visualised with the mitotic marker anti-pH3. For the reasons given above, in my opinion pH3 is a much better marker of proliferating cells than EdU. Thus, no further response ought to be required from the authors.

Referee 1's analysis of the authors response to Reviewer 2:

In my view, most of authors' responses seem appropriate.

There is one remaining concern, which is the quantification of TUNEL data, which Reviewer 2 considered essential but the authors did not do. I agree with the Reviewer that without the quantification, the data lack meaning. The authors could either quantify the data or remove the TUNEL data and the claims.

On the whole, in my opinion it is an interesting paper and a valuable contribution to understanding responses to injury in *Drosophila*. This will open further opportunities to investigate injury, regeneration and repair using the powerful genetics of *Drosophila*.

Point-by-point responses to Reviewer 1

Below, we have addressed the remaining points raised by reviewer 1 by improving the clarity of the MS text and extending and optimizing the Figure panels. Moreover, we performed additional data analyses to consolidate presented results.

Referee #1:

Reviewer 1 Revision of revised version:

The authors have made a good effort to improve the manuscript. I am satisfied by their response to the criticisms I had raised, as - on the whole - the new experiments, quantifications and analyses resolved those criticisms. I am satisfied that no further experimental revisions are required to enable publication of the work.

We thank reviewer 1 for the positive feed-back on our revisions. Moreover, we highly appreciate the extra time spent to review numerous experiments that we performed to respond to other reviewers.

The authors could pay some attention to some minor points below, when polishing the final version of the text.

On the contribution of duox in neural stem cell progenitors to the injury response, the authors wrote:

"The contribution of duox function in progenitors is an interesting point to address as repo-Gal4-driven duox-RNAi does not completely suppress injury induced proliferation. We tested this possibility, by conditional knock-down of duox with dpnT2A-Gal4; tubGal80ts. Unlike activation in glia, the results did not reveal a similar contribution of Duox function in neural progenitors. These results are shown in EV2M. We therefore consider it likely that the remaining proliferation may stem from a minor role of progenitors and from glial subsets that show no or low expression of repo in the adult OL (Ferreira and Desplan, 2023). We are discussing this possibility in the text on page.7, 1st paragraph".

Reviewer's response:

The additional experiment in EV2M for the revised version is nice, although the images on which the graph data are based ought to be provided. The data in EV2M show that upon injury, the number of pH3+ mitotic cells in the brain increased in controls (lacZ), and there was no significant difference between injured controls and injured brains with duoxRNAi in dpn+ NSCs. The authors interpret that NSC do not contribute to duox-induced proliferation after injury. However, an alternative interpretation is that since presumably there are more glial cells than Dpn+ progenitors, any effect by Dpn+ cells would be diluted. Furthermore, when comparing tubGAL80ts, dpn>duoxRNAi injured vs not-injured, there was no significant difference either, meaning that duox is required in NSC for injury-dependent mitosis. If duox were not required for injury-dependent proliferation of NSCs, then overall mitosis in injured duoxRNAi samples should have been significantly different from that of non-injured duoxRNAi controls. In fact, when they knocked-down duox in glia, there was a significant difference between injured vs

non-injured *duox*RNAi samples (Figure 2T), which had prompted the comment (raised by two reviewers) that glia could not be the only source of *duox* after injury. Thus, the authors' interpretation of data - ie that glia are required for *duox*-dependent injury-induced proliferation whereas *dpn*⁺ cells are not - is not consistent with the data shown. Instead, the data show that both glia and *dpn*⁺ cells are involved in *duox*-dependent cell proliferation after injury. This makes more sense also with what is known about how ROS induces proliferation of NSC and with previous published work by these same authors that NSCs proliferate after injury in *Drosophila*.

The conclusion lacked clarity, as we mainly intended to stress that a cell autonomous function of *duox* in progenitors may not significantly contribute to proliferation. ROS produced by the more numerous glia (from glial *Duox*) can stimulate NSC proliferation also in a cell non-autonomous manner, which is consistent with previous findings and NSC studies and which we also present in Figure 4U.

It is indeed the case that there was no increase in cell divisions in injured *dpn*>*duox* RNAi brains compared to the non-injured conditions. The fact that injury-induced proliferation in *dpn*>*duox* RNAi OLs was still not significantly different from the injured controls, makes the result more difficult to interpret. We have modified the conclusions as follows, as they lacked some clarity.

*“Duox suppression in progenitors did not significantly reduce injury-induced cell divisions compared to control flies. In turn, non-injured and injured *dpn*>*duox* RNAi OLs showed similar mitotic counts suggesting that Duox function may be needed in progenitors to support their proliferation (Figure EV2N-Q). However, it is likely that ROS released from more abundant glial cells can also considerably stimulate progenitor activation in a cell non-autonomous manner, when Duox function is suppressed in progenitors.”* (page 7, first paragraph)

The authors response including additional data and analyses to my points 2, 3 and 4 are excellent, thank you.

Addressed.

However, I am afraid the authors did not resolve point 5 on perma-twin-clones. Unfortunately, twin-spot MARCM (Yu et al 2009) and Perma-Twin (Fernandez-Hernandez et al, 2013) clones based on twin-spot MARCM with actin driven UAS-FLP - both methods - produce abundant artefacts that prevent lineage tracing. The fact that some authors continue to use these methods does not make artefacts disappear nor the science any better. Rather, it is worrying if the community perpetuates artefacts. For example, in Figure 3E,F,H,I the number of RFP⁺ and GFP⁺ cells differs greatly, with many more red cells than green cells (their quantifications ought to have been given separately in the graphs). MARCM clones require cell proliferation to occur. Generally, glial cells divide symmetrically, and if glial cells were the major contributors to injury-induced proliferation as this paper proposes, then the resulting twin red and green clones should contain equal number of glia cells, clustering together as glia normally do, and would be expected to have characteristic glial shape. But the images do not show such clones. Furthermore, the controls Figure 3E,H differ greatly. Potentially, the reason these cell division do not result in the expected progeny cells and clones

is because the micro-RNAs in twin-spot MARCM do not cause reliable, stable, persistent knock-down of RFP and GFP over time, and GFP and RFP are expressed regardless of lineage.

The effects of NAC and VitC are significant, so if they are to be kept, then Figure 3E-J should be moved to supplementary figures, GFP and RFP cell counts should be provided separately, and an explanation of the caveats of the method should be included within the main text. Having said that, the manuscript contains abundant, high-quality data, thus I strongly advise removing the perma twin MARCM data (Figure 3E-J). Removing these data (Figure 3E-J) would improve the quality of the paper, the science and the consistency of the conclusions.

We agree that novel mitotic-dependent labelling and tracing tools would be very valuable for the community as the twin-spot system has some drawbacks. Labelling in non-injured brain may arise from physiologic divisions or potentially from background due to non-reliable knock-down of GFP/RFP by the micro-RNAs in twin-spot MARCM as mentioned. If that should be the case, the twin-spot flies would also show some labelling independent of a flippase source and the analyses may allow to better tell apart any background signal in the future.

We have been working in the past two years on a system where flippase expression is controlled by elements of the QF-QUAS system, whereas the monitoring of the clones relies on classic MARCM independent of twin-spot, which may show more independent control to assess proliferation, although only one daughter fate can be traced.

To point to potential drawbacks of the system, we have incorporated comments on the twin-spot MARCM in the discussion of Figure 3 (page 8, third paragraph). Nevertheless, the injury-induced increased labelling of cells is significantly above the markage obtained with the induced system without injury. We kept the main experiment in Figure 3, added new control images of non-injured brains (Fig.3E,G) and moved further images and quantifications (prev. Figs. 3H-J to the supplement (new Figure S3A-C), in line with the notion that data of this approach is more complementary.

The main conclusion – that ROS stimulate proliferation is consistent with analyses in Figure 2 and we emphasized that conclusions are taken in conjunction with other methods. The text regarding the vitamin experiments was also adapted.

Additionally, we have increased the RFP signal in the panels. The flies tolerate the membrane-targeted RFP of the twin-spot much less than membrane GFP, which represents a challenge for imaging and the stocks need to be frequently rebuilt to achieve high signal, as expression decreases over time.

In Figure 3, we use perma-twin to detect injury-dependent cell divisions as MARCM clones “require cell proliferation to occur”. No conclusions are drawn on differential contribution of glia versus progenitors (tracing of different lineages) as a general *actin* driver is used (any dividing cell would acquire labelling). We do not propose that glia are “the major contributors to injury-induced proliferation” in the MS, but state that glia are the major “source” of oxidative stress. The extracellularly released ROS by glial Duox may stimulate proliferation in nearby cells, including progenitors (e.g. Figure 4U).

Minor:

Figure 1N: figure legend needs detail on what the Y-axis values mean in this graph. The y axis refers to normalized read counts. We have added this to the Figure legend.

Figure 3G,J GFP and RFP cell numbers should be given separately. Also, images from non-injured controls must be provided.

We have modified Figure 3 and have included images from non-injured controls (Figure 3E and G). For these experiments we used total labelled cells as a read-out.

Figure 2T, Figure 4S and Figure EV3, Figure EV2M image data should be provided.

We have added further image panels regarding the PH3 read-out shown in Figure 2S for experiments with *dpnT2A-Gal80ts* (previous Figure EV2M). They are shown in new Fig S2N-P. As regarded as valuable addition by the reviewer, we also added the close-up of cell nuclei with PH3 signal in previous Figure for Rev 3 as panels in now Fig. EV2M.

In the Methods section, the description of conditional expression in experiments using tubGal80ts, including the perma-twin MARCM, is missing. These methods were used for data in Figures 2,3,4. For example, at what temperature were flies raised, when were they shifted to what temperature, and were they kept at that temperature until dissection (and any variation across experiments)?

Thank you for spotting that information on the specific temperatures used was missing. We have added the information to the methods.

Statistical analysis has improved, but there is still room for improvement. For example, Figure 2D legend says Kruskal Wallis and Dunn test were used. However, Dunn test is to a fixed control (ie the same control for all comparisons), and instead, the graph shows comparisons of everything against everything.

Figures 2K, S, T seem to require Two Way ANOVA instead. I recommend the authors seek expert advice. Most likely it will not make a difference to the conclusions, as overall data are clear and sample sizes large enough.

Figures 2S, 2T: Although the used Kruskal-Wallis ANOVA had determined non-significant differences for two interactions shown in Fig. 2S and ST, the errors bars in photoshop had been wrongly labelled with**, which has been overlooked and we appreciate the comments that helped to correct these errors.

We subsequently also performed Two Way ANOVA on log-transformed data, followed by Tukey's multiple comparison test for data in Figure 2S and 2T based on initial results of Levene's test of data (variance).

The Kruskal-Wallis ANOVA and Two Way ANOVA gave similar results regarding significance, with the 2-way ANOVA being able to extract slightly higher significance (** instead of ***) in a few cases. All p values and tests have been added to the legends. We further applied 2-way ANOVA with Tukey's multiple comparison test for Figure EV2Q (previous **EV2M**), which led to the same result regarding significance.

Figure 2K, 2E. We have analyzed the two time points separately and performed paired t-tests as they represent different assays with their cognate control, which is the correct way to analyze them. We have introduced a divider to separate the two experiments, which were plotted together to save space.

Figure 2D: Yes, we applied and labelled the graph according to Dunn's correction, comparing injured to aged brains and non-injured to injured etc. (everything against everything). For comparisons to a fixed control, Dunnett's correction is commonly applied.

Requested feedback on the authors' response to Reviewer 3's review:
Reviewer 3 comments are included within "..".

(1) "In the first chapter, authors show that JNK is activated in injured brain and this activation is accompanied by the production of ROS and partial lipid peroxidation. Control uninjured brain with TRE-GFP expression is lacking. Authors should provide evidence that JNK is being induced by other means (eg. pucker expression)." This was a valid point, and the authors responded appropriately, providing the required image and data showing that the JNK target MMNP1 is up-regulated upon injury.

"Sentence of last para of pg 4 on the functional role of ROS in lipid peroxidation should be validated with functional data".

It is unclear what further evidence Reviewer 3 expected here. Based on this ambiguity, it would appear the authors responded appropriately.

(1) "pH3 staining: higher mag data should shown to demonstrate that these puncta are real mitotic figures"

This was a valid point. The Figure for Reviewer 3 is good and could be included in the paper as a supplementary figure, together with similar images for the other genotypes too (eg in Figure 2P,Q,R). The larger images in the revised version are also nicer than the original, as it is clearer to see increase in pH3+ spots at the lesion site.

We have incorporated higher magnification images of PH3 signal into EV2M.

"EdU incorporation is also a must in these experiments".

I disagree with this reviewer. EdU (like BrdU) label cells in S-phase. However, cells can arrest in G2, so going through S phase does not guarantee that cells will divide. In fact, EdU and BrdU are also used as evidence for polyploidy. EdU and BrdU labelling have been used in the past a lot as evidence of cell cycling because they are considerably easier to do experimentally than using mitotic markers, because S-phase lasts much longer than mitosis. Furthermore, EdU and BrdU intercalate into DNA and the label remains even after S-phase has finished and throughout subsequent divisions. Anti-pH3 is a mitotic marker instead, and more accurate evidence of cells undergoing cell division. Thus, in my view the authors' response was appropriate.

"Mito-Timer staining: high mag to see the organelles and the GFP/RFP ratio should be shown"

This was a fair point. The authors have improved the presentation of these data, both images and quantification.

"Why is there no GFP staining in panels 2A-C?"

This was a valid point. The authors have responded appropriately, by showing the split channels.

"Memb localization of Duox in 2R-T is indeed of improvement"

This was a critical point for the authors' interpretation. The images provided in Figure EV2N are not convincing, and the genotype does not seem right. If it is tubGAL4, why is expression only switched on in clusters of cells, shouldn't it come on in all cells? In any case the images are not convincing because the red signal (duox-mruby2) does not coincide with the green signal (E-cadherin) at membranes. Figure 2U-W, present in the original submission, was better than the response. Thus, this point was not improved.

We have removed the images with the E-cadherin staining as we agree that they were not of good quality.

(2) "In the 3rd chapter, authors show that glial ROS (Duox activity) is required for injury driven regeneration. The perma-twin labeling is also in need of improvement. Too many GFP, very low number of RFPs. What about twin clones? High mag should be shown and EdU labeling is a must".

This was a valid point. Although I do not agree that EdU is a must for the reasons given above, I do agree with the criticisms of the perma twin clones, which I had also raised. The authors have not resolved these criticisms.

We have addressed these questions above and pointed out drawbacks in the MS.

(3) "In the 4th figure, authors show that Calcium influx as a result of injury induces Duxo-driven ROS. Functional data to conclude this are scarce."

This was a valid point and was addressed appropriately by the authors.

"The role of JNK in regeneration should be validated by EdU experiments".

This criticism is not valid. The authors had already shown in the original submission that loss of function of puc prevented injury-induced cell proliferation, as visualised with the mitotic marker anti-pH3. For the reasons given above, in my opinion pH3 is a much better marker of proliferating cells than EdU. Thus, not further response ought to be required from the authors.

Referee 1's analysis of the authors response to Reviewer 2:

In my view, most of authors' responses seem appropriate.

There is one remaining concern, which is the quantification of TUNEL data, which Reviewer 2 considered essential but the authors did not do. I agree with the Reviewer that without the quantification, the data lack meaning. The authors could either quantify the data or remove the TUNEL data and the claims.

The initial phrasing of the experiment may have suggested that we wanted to address cell death depending on *duox* function. In the panels shown, TUNEL serves to delineate the lesion site. The effect of *duox* suppression on *gstD*-GFP was very visual and we wanted to show that this difference is not due to the fact, that one OL suffered a much smaller lesion. The projection of all apoptotic corpses shows in fact that the *duox* RNAi brain showed even more cell death. Quantifications are always informative, but the question of *duox*'s role in apoptosis represented a different aspect, that was not central to this study here.

On the whole, in my opinion it is an interesting paper and a valuable contribution to understanding responses to injury in *Drosophila*. This will open further opportunities to investigate injury, regeneration and repair using the powerful genetics of *Drosophila*.

We thank reviewer1 for the further suggestions to polish the MS and the thorough and continued evaluation of the presented experiments. We appreciate the statement that this is an interesting paper and valuable contribution to understand and further dissect injury responses.

The incorporation of the final revisions has refined the presentation and enhanced rigor and quality of the manuscript and we believe the report is now ready for publication.

Dr. Christa Rhiner
Champalimaud Foundation
Champalimaud Research
Lisbon
Portugal

Dear Christa,

I am very pleased to accept your manuscript for publication in the next available issue of EMBO reports. Thank you for your contribution to our journal.

You may qualify for financial assistance for your publication charges - either via a Springer Nature fully open access agreement or an EMBO initiative. Check your eligibility: <https://link.springer.com/journal/44319/how-to-publish-with-us>

>>> Please note that it is EMBO Reports policy for the transcript of the editorial process (containing referee reports and your response letter) to be published as an online supplement to each paper. If you do NOT want this, you will need to inform the Editorial Office via email immediately. More information is available here: <https://link.springer.com/partners/embo-press/editorial-policies#Peer%20review>